# Improved Sample Complexity For Diffusion Model Training Without Empirical Risk Minimizer Access

**Mudit Gaur**  *mgaur@purdue.edu*
*Purdue University*

**Prashant Trivedi**  *iitb.pt@gmail.com*
*University of Central Florida*

**Sasidhar Kunapuli**  *sasidhar.kunapuli@gmail.com*
*Independent Researcher*

**Amrit Singh Bedi**  *amritbedi@ucf.edu*
*University of Central Florida*

**Vaneet Aggarwal**  *vaneet@purdue.edu*
*Purdue University*

**Reviewed on OpenReview:** *https://openreview.net/forum?id=CFdNqqlqOv*

## Abstract

Diffusion models have demonstrated state-of-the-art performance across vision, language, and scientific domains. Despite their empirical success, prior theoretical analyses of the sample complexity suffer from poor scaling with input data dimension or rely on unrealistic assumptions such as access to exact empirical risk minimizers. In this work, we provide a principled analysis of score estimation, establishing a sample complexity bound of $\mathcal{O}(\epsilon^{-4})$. Our approach leverages a structured decomposition of the score estimation error into statistical, approximation, and optimization errors, enabling us to eliminate the exponential dependence on neural network parameters that arises in prior analyses. It is the first such result that achieves sample complexity bounds without assuming access to the empirical risk minimizer of score function estimation loss.

## 1 Introduction

Diffusion models have emerged as a powerful class of generative models, achieving impressive performance across tasks such as image synthesis, molecular design, and audio generation. Central to the training of these models is the estimation of the *score function*, which characterizes the reverse-time dynamics in the diffusion process. Diffusion models are widely adopted in computer vision and audio generation tasks (Ulhaq & Akhtar, 2022; Bansal et al., 2023), text generation (Li et al., 2022), sequential data modeling (Tashiro et al., 2021), reinforcement learning and control (Zhu et al., 2023), and life sciences (Jing et al., 2022; Malusare & Aggarwal, 2024). For a more comprehensive exposition of applications, we refer readers to survey paper (Chen et al., 2024).

While diffusion models exhibit strong empirical performance, understanding their sample complexity is essential to guarantee their efficiency, generalization, and scalability, enabling high-quality generation with minimal data in real-world, resource-constrained scenarios. Some of the key works studying the sample complexity are summarized in Table 1. A key limitation of sample complexity analyses of diffusion models done thus far is the lack of the presence of finite-time sample complexity results under reasonable assumptions. This makes the theoretical analysis of diffusion models fall short of other machine learning areas such as reinforcement Learning (Kumar et al., 2023; Gaur et al., 2024), bi-level-optimization (Grazzi et al., 2023;

| Reference | Sample Complexity | Empirical Risk Minimizer Assumption |
|---|---|---|
| (Zhang et al., 2024) | $\tilde{O}\left(\epsilon^{-d}\right)$ | Yes |
| (Wibisono et al., 2024) | $\tilde{O}\left(\epsilon^{-(d)}\right)$ | Yes |
| (Oko et al., 2023) | $\tilde{O}\left(\epsilon^{-O(d)}\right)$ | Yes |
| (Chen et al., 2023) | $\tilde{O}\left(\epsilon^{-O(d)}\right)$ | Yes |
| (Gupta et al., 2024) | $\tilde{O}\left(\epsilon^{-5}\right)$ | Yes |
| **This work** | $\tilde{O}(\epsilon^{-4})$ | No |

Table 1: Summary of sample complexity results for diffusion models, assuming no upper bound on score estimation error.

Gaur et al., 2025) and graphical models (Fattahi et al., 2019; Tran et al., 2019). In this work we aim to bridge that gap and obtain a sample complexity results on the same footing as results from the aforementioned areas. The iteration complexity or convergence has been studied in Li et al. (2024b); Benton et al. (2024); Li & Yan (2024); Huang et al. (2026); Dou et al. (2024); Liang et al. (2025a;b), while they assume bounded score estimates thus not providing the sample complexity which requires estimating the score function.

We note that works such as (Zhang et al., 2024; Wibisono et al., 2024; Oko et al., 2023; Chen et al., 2023) have sample complexity results that depend exponentially on the data dimension, making the result less useful in high-dimensional settings. Recently, Gupta et al. (2024) improved upon this by obtaining $\tilde{\mathcal{O}}(\epsilon^{-5})$ sample complexity without exponential dependence on data dimension. . However, this work assumes access to the empirical risk minimizer (ERM) of the score estimation loss, a significant restriction that was explicitly highlighted as an open problem in Gupta et al. (2024) itself. While this assumption is present in all prior works, it is an unrealistic assumption regardless.

In this paper, we do not make these assumptions and establish an improved state-of-the-art sample complexity bound of $\tilde{\mathcal{O}}(\epsilon^{-4})$. This represents a key step toward bridging the gap between the theory and practice of diffusion models. Specifically, we address the following fundamental question.

*How many samples are required for a sufficiently expressive neural network to estimate the score function well enough to generate high-quality samples using a DDPM algorithm?*

Our analysis directly connects the quality of the learned score function to the total variation distance between the generated and target distributions, offering more interpretable and practically relevant guarantees. Additionally, our work accounts for the unavailability of the empirical risk minimizer. Using our novel analysis of the score estimation error, we obtain the sample complexity bounds without exponential dependence on the data-dimension. Our principled analysis accounts for the statistical and optimization errors while not assuming access to the empirical risk minimizer of the score estimation loss, and achieves state-of-the-art sample complexity bounds.

The statistical error occurs due to the finite sample size used to obtain the score estimate. Existing methods used to bound statistical errors assume bounded loss functions, which is not true in the case of diffusion models. We thus use a novel analysis that uses the conditional normality of the score function to obtain upper bounds for the statistical error.

Finally, the optimization error occurs due to a finite number of SGD steps during the estimation of the score function. It is precisely the error that was not accounted for in the previous works due to the assumption that they have access to the empirical risk minimizer. We use the quadratic growth property implied by the Polyak-Łojasiewicz (PL) condition assumed in Assumption 2 and a novel recursive analysis of the error at each stochastic gradient descent (SGD) step to upper bound this error. While the PL assumption is restrictive and not globally applicable, PL is essentially the weakest standard assumption that yields global convergence rates strong enough to close a finite-sample optimization analysis (Liu et al., 2022). Our goal is to remove the assumption of access to an empirical risk minimizer and instead provide an explicit, finite-time SGD guarantee that appears as an optimization error term in the three-way decomposition. To achieve this,

some form of global error-gradient relationship is required. Among commonly used assumptions, the PL inequality is a canonical and comparatively mild choice: it is compatible with nonconvex landscapes and is significantly weaker than convexity, while still enabling global convergence guarantees for SGD.

The main contributions of our work are summarized as:

- **Finite time sample complexity bounds.** We derive state-of-the-art sample complexity bound of $\widetilde{\mathcal{O}}(\epsilon^{-4})$ for score-based diffusion models, without exponential dependence on the data dimension or neural network parameters. Our analysis avoids the unrealistic assumptions used in prior works such as access to an empirical loss minimizer.

- **Principled error decomposition.** We propose a structured decomposition of the score estimation error into approximation, statistical, and optimization components, enabling the characterization of how each factor contributes to sample complexity.

It is to be noted that the sample complexity obtained in our work is exponential with respect to neural network parameters for our work in contrast with works such as (Gupta et al., 2024), where the sample complexity is polynomial with respect to the neural network parameters. We leave for future work results which obtain upper bounds without the ERM assumption which are polynomial in the neural network parameters.

**Unconditional and Conditional Diffusion Models.** Diffusion models have emerged as leading frameworks across vision, audio, and scientific domains. Foundational works such as Sohl-Dickstein et al. (2015) and Ho et al. (2020) introduced and refined Denoising Diffusion Probabilistic Models (DDPMs), enabling high-quality sample generation. Subsequent advances include improved noise schedules (Nichol & Dhariwal, 2021), score-based SDE formulations (Song et al., 2021), and efficient latent-space generation via Latent Diffusion Models (LDMs) (Rombach et al., 2022). Conditional diffusion models extend these techniques for guided generation tasks, with applications in time-series (Tashiro et al., 2021), speech (Huang et al., 2022), and medical imaging (Dorjsembe et al., 2023). Conditioning mechanisms range from classifier-based (Dhariwal & Nichol, 2021) to classifier-free guidance (Ho & Salimans, 2022), which enabled text-to-image models like Imagen (Saharia et al., 2022) and Stable Diffusion (Rombach et al., 2022). Recent innovations focus on adaptive control (Castillo et al., 2025), compositionality (Liu et al., 2023), and multi-modal conditioning (Avrahami et al., 2022).

**Related Works:** Despite the empirical success of diffusion models, theoretical understanding regarding the sample complexity remains limited. Assuming accurate score estimates, authors in (Chen et al., 2022) showed that score-based generative models can efficiently sample from a sub-Gaussian data distribution. Assuming a bounded score function, iteration complexity bounds have been extensively studied in recent works (Li et al., 2024b; Benton et al., 2024; Li & Yan, 2024; Huang et al., 2026; Dou et al., 2024; Liang et al., 2025a;b). Some works, such as Zhang & Pilanci (2024), establish iteration complexity for score matching. In particular, (Benton et al., 2024; Li et al., 2024b) establishes iteration complexity guarantees for DDPM algorithms. Several studies propose accelerated denoising schedules to improve these rates (Li et al., 2024a; Liang et al., 2025b; Dou et al., 2024). Additionally, improved convergence rates under low-dimensional data assumptions are demonstrated in (Li & Yan, 2024; Huang et al., 2026; Liang et al., 2025a). In contrast to these works, our analysis addresses the sample complexity of score-based generative models, where the errors introduced by the neural network approximation, data sampling, and optimization are also accounted for.

Sample complexity bounds for diffusion models have been studied via diffusion SDEs under smoothness and spectral assumptions in (Chen et al., 2023; Zhang et al., 2024; Wibisono et al., 2024; Oko et al., 2023). However, these bounds are exponential in the data dimension. Further, authors of (Gupta et al., 2024) use the quantile-based approach to get the sample complexity bounds. In this work, we further improve on these guarantees. The detailed comparison of sample complexities with the key approaches mentioned above is provided in Table 1.

## 2 Preliminaries and Problem Formulation

We begin by outlining the theoretical basis of score-based diffusion models. In particular, we adopt the continuous-time stochastic differential equation (SDE) framework, which provides a principled basis for modeling the generative process. We then outline its practical discretization and formally define our problem.

Score-based generative models enable sampling from a complex distribution $p_0$ by learning to reverse a noise-adding diffusion process. This approach introduces a continuous-time stochastic process that incrementally perturbs the data distribution into a tractable distribution (typically Gaussian), and then seeks to reverse that transformation.

A canonical forward process used in diffusion models is the *Ornstein–Uhlenbeck (OU) process* (Øksendal, 2003), defined by the following SDE

$$dx_t = -x_t dt + \sqrt{2} dB_t, \quad x_0 \sim p_0, x \subset \mathbb{R}^d, \tag{1}$$

where $B_t$ denotes standard Brownian motion. The solution of this SDE in closed form is given by

$$x_t \sim e^{-t} x_0 + \sqrt{1 - e^{-2t}} \epsilon, \quad \text{with } \epsilon \sim \mathcal{N}(0, I). \tag{2}$$

As $t \to \infty$, the process converges to the stationary distribution $\mathcal{N}(0, I)$. Let $p_t$ denote the marginal distribution of $x_t$. This defines a continuous-time smoothing of the data distribution, where $p_t$ becomes increasingly Gaussian over time.

The reverse process is typically achieved using stochastic time-reversal theory (Anderson, 1982), which yields a corresponding reverse-time SDE as follows.

$$dx_{T-t} = (x_{T-t} + 2\nabla \log p_{T-t}(x_{T-t})) \, dt + \sqrt{2} \, dB_t, \tag{3}$$

where $\nabla \log p_t(x)$ is known as the *score function* of the distribution $p_t$. Simulating this reverse process starting from $x_T \sim p_T \approx \mathcal{N}(0, I)$ yields approximate samples from the original distribution $p_0$. This motivates a sampling strategy where we begin from $x_T \sim \mathcal{N}(0, I)$ for sufficiently large $T$, and then integrate the reverse SDE backward to $t = 0$ using estimated score functions. In practice, the backward process is run up to a fixed time point $t_0$ known as the *early stopping time* and not $t = 0$. This is done in order to improve performance and training speed (Lyu et al., 2022; Favero et al., 2025).

The continuous-time reverse SDE (Equation 3) is discretized over a finite sequence of times $0 < t_0 < t_1 < \cdots, t_k, \cdots < t_K = (T - \kappa) < T$. The score function $s_t(x) := \nabla \log p_t(x)$ is approximated at these discrete points using a learned estimator $\hat{s}_{t_k}$. This discretization underlies the DDPM framework (Ho et al., 2020), where the reverse process is implemented by iteratively denoising the sample using the estimated scores at each time step. The detailed procedure is provided in Algorithm 1 in the Appendix B. We employ stochastic gradient descent (SGD) to learn the score function at each $t_k$, using either a constant learning rate, as justified in our analysis later.

**Problem formulation:** Let the score function at time $t_k$ be approximated using a parameterized family of neural networks $\mathcal{F}_\Theta^k = \{s_\theta : \theta \in \Theta_k\}$, where each $s_\theta : \mathbb{R}^d \times [0, T] \to \mathbb{R}^d$ is represented by a neural network of depth $D$ and width $W$ with smooth activation functions. Given $n_k$ i.i.d. samples $\{x_i\}_{i=1}^k$ from the data distribution $p_{t_k}$, the score network is trained by minimizing the following loss:

$$\mathcal{L}_k(\theta) := \mathbb{E}_{x \sim p_{t_k}} \left[ \|s_\theta(x, t_k) - \nabla \log p_{t_k}(x)\|^2 \right]. \tag{4}$$

**Objective.** Our goal is to quantify how well the learned generative model $\hat{p}_{t_0}$ approximates the true data distribution $p$ in terms of total variation (TV) distance. Specifically, we aim to show the number of samples needed so that with high probability, the TV distance $\text{TV}(p_{t_0}, \hat{p}_{t_0})$ is bounded by $\mathcal{O}(\epsilon)$, where $\epsilon$ is the $L^2$ estimation error of the score function. This reduces the generative performance analysis to establishing tight sample complexity bounds on the score estimation error. We additionally define the following probability

distributions:

$$p_{t_0} := \textit{Distribution obtained after backward process till time } t_0 \textit{ steps starting form } p_T$$

$$p_{t_0}^{dis} := \textit{Distribution obtained by backward process till time } t_0 \textit{ starting from } p_T$$
$$\textit{at discretized time steps}$$

$$\tilde{p}_{t_0} := \textit{Distribution obtained by backward process till time } t_0 \textit{ starting from } p_T$$
$$\textit{at discretized time steps using the estimated score functions}$$

$$\hat{p}_{t_0} := \textit{Distribution obtained by backward process till time } t_0 \textit{ starting from } \mathcal{N}(0, I)$$
$$\textit{at discretized time steps using the estimated score functions}$$

where $t_0$ denotes the early stopping time.

## 3 Sample Complexity of Diffusion Models

In this section, we derive explicit sample complexity bounds for diffusion-based generative models. By leveraging tools from stochastic optimization and statistical learning theory, we provide bounds on the number of data samples required to accurately estimate the time-dependent score function $s_t(x) := \nabla \log p_t(x)$ across the forward diffusion process. Note that accurate score estimation is critical for ensuring high-quality generation while sampling through the reverse-time SDE.

We first state the assumptions required throughout this work.

**Assumption 1** (Bounded Second Moment Data Distribution.)**.** *The data distribution $p_0$ of the data variable $x_0$ has an absolutely continuous CDF, is supported on a continuous set $\Gamma \in \mathbb{R}^d$, and there exists a constant $0 < C_1 < \infty$ such that $\mathbb{E}(||x_0||^2) \leq C_1$.*

Works such as (Gupta et al., 2024; Wibisono et al., 2024) also assume a second bounded moment as we have done here. (Zhang et al., 2024) assumes a sub-gaussian assumption while works such as (Chen et al., 2022; Oko et al., 2023), assume that the data distribution is supported on a bounded set, thereby excluding commonly encountered distributions such as Gaussian and sub-Gaussian families. In contrast, our analysis only requires the data distribution to have a finite second moment, making our results applicable to a significantly broader class of distributions. Note that the works that assume a sub-Gaussian assumption or bounded assumption are able to account for the error incurred due to a finite stopping time, while works such as (Gupta et al., 2024) and this work do not. In this work, we show that if we assume a sub-Gaussian data distribution, we can also account for the error due to a finite stopping time.

**Assumption 2** (Polyak - Łojasiewicz (PL) condition.)**.** *The loss $\mathcal{L}_k(\theta)$ for all $k \in [0, K]$ satisfies the Polyak–Łojasiewicz condition, i.e., there exists a constant $\mu > 0$ such that*

$$\frac{1}{2} \|\nabla \mathcal{L}_k(\theta)\|^2 \geq \mu \left( \mathcal{L}_k(\theta) - \mathcal{L}_k(\theta^*) \right), \quad \forall\, \theta \in \Theta_k, \tag{5}$$

*where $\theta^* = \arg\min_{\theta \in \Theta_k} \mathcal{L}(\theta)$ denotes the global minimizer of the population loss.*

The Polyak-Łojasiewicz (PL) condition is significantly weaker than strong convexity and is known to hold in many non-convex settings, including overparameterized neural networks trained with mean squared error losses (Liu et al., 2022). Prior works such as (Gupta et al., 2024) and (Block et al., 2020) implicitly assume access to an exact empirical risk minimizer (ERM) for score function estimation, as reflected in their sample complexity analyses (see Assumption A2 in (Gupta et al., 2024) and the definition of $\hat{f}$ in Theorem 13 of (Block et al., 2020)). This assumption, however, introduces a major limitation for practical implementations, where exact ERM is not attainable.

In contrast, the PL condition allows us to derive sample complexity bounds under realistic optimization dynamics, without requiring exact ERM solutions. To our knowledge, this is the first theoretical analysis of score-based generative models that explicitly accounts for inexact optimization, addressing a key gap in existing literature. Additionally, we establish convergence guarantees with both constant and decreasing step sizes.

**Assumption 3** (Approximation error of the Class of Neural Networks)**.** *For all $t \in [0, T]$, there exists a neural network parameter $\theta \in \Theta_k$ such that*

$$\mathbb{E}_{x \sim p_t} ||s_\theta(x, t) - \nabla \log p_t(x)||^2 \leq \epsilon_{approx} \tag{6}$$

This error is independent of the sampling algorithm, and describes the error due to neural network parametrization. In learning theory, it is common to treat the *approximation error* of a model class as a constant so that analyzes can focus on the estimation/ optimization terms dependent on the sample. This convention appears in standard excess-risk decompositions for fixed hypothesis classes (Shalev-Shwartz & Ben-David, 2014). In PAC-Bayesian analyses, approximation errors are denoted by a constant once the class is fixed (Mai, 2025). In (NTK/RKHS) analyses of neural networks, where it is assumed the target function lies in, or is well approximated by the specified function class, the misspecification error is represented as a constant term (Bing et al., 2025). In reinforcement learning algorithm analysis such as policy gradient, a task-dependent "inherent Bellman" or function-approximation error that remains constant while deriving performance rates (Mondal & Aggarwal, 2024; Fu et al., 2021; Gaur et al., 2024; Ganesh et al., 2025). Note that in Gupta et al. (2024), Assumption A.2 states that the error in estimating the loss function is 'sufficiently small'. In practice, this assumption is used to make the score function estimation error arbitrarily small, as is done in Theorem C.3, where it is stated that the there exists a neural network such that the error in estimating the loss function is $\mathcal{O}(\epsilon^3)$. Thus, this is a stronger assumption as compared to our Assumption 3.

Note that in certain works, such as (Jiao et al., 2023), it is shown that the network size has to be exponential in data dimension in order to achieve a small approximation error. However, in practice, that would require an impractically large neural network size. In practice neural network size is of the same order as the data dimension. Thus for a fixed neural network size that we assume in this work, it makes sense to assume the approximation error as a constant.

**Assumption 4** (Smoothness and bounded gradient variance of the score loss.)**.** *For all $k \in [0, K]$, the population loss $\mathcal{L}_k(\theta)$ is $\kappa$-smooth with respect to the parameters $\theta$, i.e., for all $\theta, \theta' \in \Theta_k$*

$$||\nabla \mathcal{L}_k(\theta) - \nabla \mathcal{L}_k(\theta')|| \leq \kappa ||\theta - \theta'||. \tag{7}$$

*We assume that the estimators of the gradients $\nabla \mathcal{L}_k(\theta)$ have bounded variance.*

$$\mathbb{E}||\nabla \widehat{\mathcal{L}}_k(\theta) - \nabla \mathcal{L}_k(\theta)||^2 \leq \sigma^2. \tag{8}$$

Together, these assumptions form a minimal yet sufficient foundation for analyzing score estimation in practice. *Smoothness* and *bounded gradient variance* implied by the sub-Gaussian assumption are mild and generally satisfied for standard neural architectures such as GELU activations. The *PL condition* has been shown to emerge in over-parameterized networks or under lazy training regimes, where the function class is expressive enough to approximate the ground-truth score function (Liu et al., 2022). Notably, these conditions are not only specific to our setting they have been widely adopted in recent works studying the optimization landscape of deep diffusion models (Salimans & Ho, 2022; Liu et al., 2022). Note that in no prior works were such assumptions stated since they assumed access to the empirical risk minimizer.

It is to be noted that assumptions 2, 3, and 4 are not included in prior works, since they assumed access to the ERM. Since we do not make that assumption, these additional assumptions are needed for obtaining upper bounds on the Total variation between the true and generated distributions.

**Theorem 1** (Total Variation Distance Bound)**.** *Let $p_{t_0}$ denote the distribution obtained by the backward process till time $t_0$ starting form $p_T$, and $\hat{p}_{t_k}(x)$ be the distribution generated by the backward process at discretized time steps $\{t_k\}$, starting from $\mathcal{N}(0, I)$ using the estimated score functions $\hat{s}_{t_k}(x)$ where $k \in [0, K]$. Let $d$ be the data dimension, and $n_k$ be the number of samples for score estimation at time step $t_k$.*

*Assume that the data distribution satisfies Assumption 1, the loss function $\mathcal{L}_k(\theta)$ satisfies Assumptions 2, 3,4 for all $k \in [0, K]$ and the learning rate for estimating $\mathcal{L}_k(\theta)$ using SGD satisfies $0 \leq \eta \leq \frac{1}{\kappa}$ for all $k \in [0, K]$. Further assume*

$$n_k = \Omega \left( W^{2D} \cdot d^2 \cdot \log \left( \frac{4K}{\delta} \right) \left( \frac{\epsilon^{-4}}{\sigma_k^{-4}} \right) \right), \tag{9}$$

*Then, with probability at least $1 - \delta$, the total variation distance between the $p_{t_0}$ and $\hat{p}_{t_0}$ satisfies*

$$TV(p_{t_0}, \hat{p}_{t_0}) \leq \mathcal{O}(\exp^{-T}) + \mathcal{O}\left(\frac{1}{\sqrt{K}}\right) + \mathcal{O}\left(\epsilon \cdot \sqrt{\left(T + \log\frac{1}{\kappa}\right)}\right) + \epsilon_{approx} \tag{10}$$

*Furthermore, by setting $T = \Omega\left(\log\left(\frac{1}{\epsilon}\right)\right), \kappa = \Omega(\epsilon)$ and $K = \Omega(\epsilon^{-2})$, we obtain*

$$\mathrm{TV}(p_{t_0}, \hat{p}_{t_0}) \leq \mathcal{O}(\epsilon) + \epsilon_{approx}, \tag{11}$$

*with probability at least $1 - \delta$.*

Theorem 1 establishes that the total variation distance between the true data distribution and the diffusion model's output can be made arbitrarily small specifically, $\tilde{\mathcal{O}}(\epsilon)$ by properly scaling model capacity and algorithmic parameters. To the best of our knowledge, these are the only known sample complexity bounds for score-based diffusion models, improving upon the prior results as discussed in the introduction without assuming access to empirical risk minimizer for the score estimation loss.

**Usage of $p_{t_0}$ instead of $p_0$ in Theorem 1**: We have shown that the estimated distribution $\hat{p}_{t_0}$ is $\mathcal{O}(\epsilon)$-close in total variation (TV) to $p_{t_0}$, where $p_{t_0}$ denotes the data distribution $p_0$ pushed forward by $t_0$ steps of the forward process. We do not claim that $\hat{p}_{t_0}$ is $\mathcal{O}(\epsilon)$-close in TV to the true data distribution $p_0$ (i.e., we do not bound $\mathrm{TV}(p_0, \hat{p}_{t_0})$), because doing so would require additional assumptions on $p_0$. For example, Fu et al. (2024) (in Lemma D.5) assumes a sub-Gaussian data distribution to show that $\mathrm{TV}(p_0, p_{t_0}) \leq \mathcal{O}\left(\sqrt{t_0}\log(1/t_0)\right)$. We also note that all other works listed in Table 1 similarly provide upper bounds on $\mathrm{TV}(p_{t_0}, \hat{p}_{t_0})$, not on $\mathrm{TV}(p_0, \hat{p}_{t_0})$.

However, it is to be noted that using the sub-Gaussian assumption, our analyis can be extended to a bound $\mathrm{TV}(p_0, \hat{p}_{t_0})$ via the triangle inequality:

$$\mathrm{TV}(p_0, \hat{p}_{t_0}) \ \leq \ \mathrm{TV}(p_0, p_{t_0}) \ + \ \mathrm{TV}(p_{t_0}, \hat{p}_{t_0}).$$

We formally present the data assumption and the resulting theorem as follows

**Assumption 5** (Sub-Gaussian Data Distribution.). *The data distribution $p_0$ of the data variable $x_0$ has an absolutely continuous CDF, is supported on a continuous set $\Gamma \in \mathbb{R}^d$, and there exists a constant $0 < C_2 < \infty$ such that for every $t \geq 0$ we have $P(|x_0| \geq t) \leq 2 \cdot \exp^{-\frac{t^2}{C_2^2}}$.*

**Theorem 2** (Total Variation Distance Bound Under Sub-Gaussian Assumption). *Assume that the data distribution satisfies Assumption 5, the loss function $\mathcal{L}_k(\theta)$ satisfies Assumptions 2 3,4 for all $k \in [0, K]$ and the learning rate satisfies for estimating $\mathcal{L}_k(\theta)$ using SGD satisfies $0 \leq \eta \leq \frac{1}{\kappa}$ for all $k \in [0, K]$. Further assume*

$$n_k = \Omega\left(W^{2D} \cdot d^2 \cdot \log\left(\frac{4K}{\delta}\right)\left(\frac{\epsilon^{-4}}{\sigma_k^{-4}}\right)\right), \tag{12}$$

*Then, with probability at least $1 - \delta$, the total variation distance between the $p_0$ and $\hat{p}_{t_0}$ satisfies*

$$TV(p_0, \hat{p}_{t_0}) \leq \mathcal{O}\left(\sqrt{t_0}\log(1/t_0)\right) + \mathcal{O}(\exp^{-T}) + \mathcal{O}\left(\frac{1}{\sqrt{K}}\right)$$
$$+ \mathcal{O}\left(\epsilon \cdot \sqrt{\left(T + \log\frac{1}{\kappa}\right)}\right) + \epsilon_{approx} \tag{13}$$

*Furthermore, by setting $t_0 = \Omega(\epsilon^2)$, $T = \Omega\left(\log\left(\frac{1}{\epsilon}\right)\right), \kappa = \Omega(\epsilon)$ and $K = \Omega(\epsilon^{-2})$, we obtain*

$$\mathrm{TV}(p_0, \hat{p}_{t_0}) \leq \mathcal{O}(\epsilon) + \epsilon_{approx}, \tag{14}$$

*with probability at least $1 - \delta$.*

**Proof of Theorem 1.**

Recall that $\hat{p}_{t_0}$ is derived via score-based sampling, so using the triangle inequality repeatedly to decompose the TV distance between the true distribution $p_{t_0}$ and $\hat{p}_{t_0}$ we obtain

$$\mathrm{TV}(p_{t_0}, \hat{p}_{t_0}) \leq \mathrm{TV}(p_{t_0}, p_{t_0}^{dis}) + \mathrm{TV}(p_{t_0}^{\mathrm{dis}}, \widetilde{p}_{t_0}) + \mathrm{TV}(\widetilde{p}_{t_0}, \hat{p}_{t_0}) \tag{15}$$

The bounds on $\mathrm{TV}(p_{t_0}, p_{t_0}^{dis})$ and $\mathrm{TV}(\widetilde{p}_{t_0}, \hat{p}_{t_0})$ follow from Lemma B.4 of (Gupta et al., 2024) and Proposition 4 of (Benton et al., 2024), respectively to get

$$\mathrm{TV}(p_{t_0}, \hat{p}_{t_0}) \leq \mathcal{O}\left(\frac{1}{\sqrt{K}}\right) + \mathrm{TV}(p_{t_0}^{\mathrm{dis}}, \widetilde{p}_{t_0}) + \mathcal{O}(\exp(-T)) \tag{16}$$

Note that we have used results from (Gupta et al., 2024) and (Benton et al., 2024) which assume a bounded second moment for the data distribution. This is satisfied by Assumption 1. Now from lemma 4, $\mathrm{TV}(p_{t_0}^{\mathrm{dis}}, \widetilde{p}_{t_0})$ is upper bounded as follows

$$\mathrm{TV}(p_{t_0}^{\mathrm{dis}}, \widetilde{p}_{t_0}) \leq \frac{1}{2}\sqrt{\sum_{k=0}^{K} E_{x \sim p_{t_k}} \left\| \hat{s}_{t_k}(x, t_k) - \nabla \log p_{t_k}(x) \right\|^2 (t_{k+1} - t_k)} \tag{17}$$

In order to upper bound $\mathrm{TV}(p_{t_0}^{\mathrm{dis}}, \widetilde{p}_{t_0})$, we denote $A(k)$ as

$$A(k) := E_{x \sim p_{t_k}} \left\| \hat{s}_{t_k}(x, t_k) - \nabla \log p_{t_k}(x) \right\|^2 \tag{18}$$

Therefore, bounding the TV distance between $p_{t_0}$ and $\hat{p}_{t_0}$ translates to bounding the cumulative error in estimating the score function at different time steps. We now focus on bounding this term. Specifically, we have that

$$\mathrm{TV}(p_{t_0}, \hat{p}_{t_0}) \leq \mathcal{O}\left(\frac{1}{\sqrt{K}}\right) + \frac{1}{2}\sqrt{\sum_{k=0}^{K} A(k) \cdot (t_{k+1} - t_k)} + \mathcal{O}(\exp(-T)) \tag{19}$$

Now, for each time step $k$, we decompose the total score estimation error, denoted by $A(k)$, into three primary components: approximation error, statistical error, and optimization error. Each of these error corresponds to a distinct aspect of learning the reverse-time score function in a diffusion model as described below.

$$\mathbb{E}_{x \sim p_{t_k}}\left[\left\|\hat{s}_{t_k}(x, t_k) - \nabla \log p_{t_k}(x)\right\|^2\right] \leq 4\underbrace{\mathbb{E}_{x \sim p_{t_k}(x)}\left[\left\|s_{t_k}^a(x, t_k) - \nabla \log p_{t_k}(x)\right\|^2\right]}_{\mathcal{E}_k^{\mathrm{approx}}}$$

$$+ 4\underbrace{\mathbb{E}_{x \sim p_{t_k}(x)}\left[\left\|s_{t_k}^a(x, t_k) - s_{t_k}^b(x, t_k)\right\|^2\right]}_{\mathcal{E}_k^{\mathrm{stat}}}$$

$$+ 4\underbrace{\mathbb{E}_{x \sim p_{t_k}(x)}\left[\left\|\hat{s}_{t_k}(x, t_k) - s_{t_k}^b(x, t_k)\right\|^2\right]}_{\mathcal{E}_k^{\mathrm{opt}}}, \tag{20}$$

where, we define the parameters

$$\theta_k^a = \arg \min_{\theta \in \Theta} \mathbb{E}_{x \sim p_{t_k}} \left[ \|s_\theta(x, t_k) - \nabla \log p_t(x, t_k)\|^2 \right], \tag{21}$$

$$\theta_k^b = \arg \min_{\theta \in \Theta} \frac{1}{n} \sum_{i=1}^n \|s_\theta(x_i, t_k) - \nabla \log p_t(x_i, t_k)\|^2 \tag{22}$$

and denote $s_{t_k}^a$ and $s_{t_k}^b$ as the estimated score functions associated with the parameters $\theta_k^a$ and $\theta_k^b$ respectively. *Approximation error* $\mathcal{E}_k^{\mathrm{approx}}$ captures the error due to the limited expressiveness of the function class $\{s_\theta\}_{\theta \in \Theta}$. The *statistical error* $\mathcal{E}_k^{\mathrm{stat}}$ is the error from using a finite sample size. Finally, the *optimization error* $\mathcal{E}_k^{\mathrm{opt}}$ is due to not reaching the global minimum during training.

One of our key contributions lies in rigorously bounding each of these error components and showing how their interplay governs the overall generative error. In particular, we derive novel bounds that explicitly capture the dependencies on sample size, neural network capacity, and optimization parameters, without any assumption on the access to the empirical risk minimizer of the score estimation loss. We formalize these results in the following lemmas. Detailed proofs are deferred to Appendices C.1, and C.2, respectively.

**Lemma 1** (Approximation Error). *$\mathcal{E}_k^{\mathrm{approx}}$ is defined as follows*

$$\mathcal{E}_k^{\mathrm{approx}} = min_{\theta \in \Theta} \mathbb{E}_{x \sim p_{t_k}} \left[ \|s_\theta(x, t_k) - \nabla \log p_t(x, t_k)\|^2 \right] \tag{23}$$

*Then, under Assumption 3 for all $k \in [0, K]$, we have*

$$\mathcal{E}_k^{\mathrm{approx}} \leq \epsilon_{approx} \tag{24}$$

This result directly follows from Assumption 3 and the definition of $\mathcal{E}_k^{\mathrm{approx}}$.

**Lemma 2** (Statistical Error). *Let $n_k$ denote the number of samples used to estimate the score function at time step $t_k$. If the data distribution satisfies the Assumption 1 and the loss function $\mathcal{L}_k(\theta)$ satisfies Assumptions 2 for all $k \in [0, K]$, then with probability at least $1 - \delta$, we have*

$$\mathcal{E}_k^{\mathrm{stat}} \leq \mathcal{O}\left( W^D \cdot d \cdot \sqrt{\frac{\log\left(\frac{2}{\delta}\right)}{n_k}} \right) \tag{25}$$

**Proof Outline** We present the outline of the proof, full details are deferred to the Appendix.

$$\mathcal{L}'_k(\theta) = \mathbb{E}_{x \sim \mu_{t_k}} \|v_\theta(x, t_k) - v_{t_k}(x)\|^2, \tag{26}$$

and

$$\widehat{\mathcal{L}}'_k(\theta) = \frac{1}{n} \sum_{i=1}^n \|v_\theta(x_i, t_k) - v_{t_k}(x_i)\|^2. \tag{27}$$

Thus, we have

$$\mathcal{L}(\theta_k^b) - \mathcal{L}_k(\theta_k^a) \leq \mathcal{L}_k(\theta_k^b) - \mathcal{L}_k(\theta_k^a) + \widehat{\mathcal{L}_k}(\theta_k^a) - \widehat{\mathcal{L}_k}(\theta_k^b), \tag{28}$$

We get the inequality by adding the term $\widehat{\mathcal{L}}(\theta_k^a) - \widehat{\mathcal{L}}(\theta_k^b)$ to the right hand side of Equation (28), this is a positive quantity since $\theta^b$ is the minimizer of $\hat{\mathcal{L}}(\theta)$, where the added term is non-negative due to the empirical optimality of $\theta_k^b$.

$$\left| \mathcal{L}_k(\theta_k^b) - \mathcal{L}(\theta_k^a) \right| \leq \underbrace{\left| \mathcal{L}(\theta_k^b) - \widehat{\mathcal{L}}(\theta_k^b) \right|}_{(I)} + \underbrace{\left| \mathcal{L}(\theta_k^a) - \widehat{\mathcal{L}}(\theta_k^a) \right|}_{(II)}. \tag{29}$$

Now, note that the terms (I) and (II) are unbounded, and thus generalization results such as Lemma 5 (Theorem 26.5 of (Shalev-Shwartz & Ben-David, 2014)), do not apply directly. Thus we define two supplementary loss functions as

$$\mathcal{L'}_k(\theta) = \mathbb{E}_{x \sim \mu_{t_k}} \|v_\theta(x, t_k) - v_{t_k}(x)\|^2 , \tag{30}$$

and

$$\widehat{\mathcal{L'}}_k(\theta) = \frac{1}{n} \sum_{i=1}^{n} \|v_\theta(x_i, t_k) - v_{t_k}(x_i)\|^2 . \tag{31}$$

where we define the functions

$$(v_t(x))_j = \begin{cases} (\nabla \log p_t(x))_j & \text{if } |\frac{x - e^{-t}x_0}{\sigma_t^2}|_j \leq \kappa \\ 0 & \text{if } |\frac{x - e^{-t}x_0}{\sigma_t^2}|_j \geq \kappa \end{cases} \tag{32}$$

and

$$(v_\theta(x, t))_j = \begin{cases} (s_\theta(x, t))_j & \text{if } |\frac{x - e^{-t}x_0}{\sigma_t^2}|_j \leq \kappa \\ 0 & \text{if } |\frac{x - e^{-t}x_0}{\sigma_t^2}|_j \geq \kappa \end{cases} \tag{33}$$

Here $(v_t(x))_j$, $(\nabla \log p_t(x))_j$, $(v_\theta(x, t))_j$ and $(s_\theta(x, t))_j$ denote the $j^{th}$ co-ordinate of $v_t(x)$, $(\nabla \log p_t(x))$, $v_\theta(x, t)$ and $s_\theta(x, t)$ respectively. Further, $|\frac{x - e^{-t}x_0}{\sigma_t^2}|_j$ denotes the $j^{th}$ co-ordinate of the $i^{th}$ sample of the score function in $\widehat{\mathcal{L}}_k(\theta)$ which is given by $\log p_t(x) = |\frac{x - e^{-t}x_0}{\sigma_t^2}|$.

Note that the functions $\mathcal{L'}_k(\theta)$ and $\widehat{\mathcal{L'}}_k(\theta)$ are bounded and thus the result from 5 (Theorem 26.5 of (Shalev-Shwartz & Ben-David, 2014)) are applicable. Additionally, we bound the error due to the truncation in the terms $\mathcal{L'}_k(\theta)$ and $\widehat{\mathcal{L'}}_k(\theta)$ using the conditional normality of the score function given the initial data distribution as well as bounded second moment of the initial data distribution. This is the component of the error that accounts for the fact that we have a finite sample size and thus we solve an empirical loss function given in (21). The proof of this lemma follows from utilizing the definitions of $s_t^a$ and $s_t^b$. Existing analyses of statistical errors, such as those given in (Shalev-Shwartz & Ben-David, 2014), only work when the loss function is bounded. This is not the case for diffusion models. Thus, we use a novel analysis that uses the conditional normality of the score function as well as the bounded second moment property of the data variable in Assumption 1 to obtain the upper bound on the statistical error. The details of the proof are given in Appendix C.1.

**Lemma 3** (Optimization Error). *Let $n_k$ be the number of samples used to estimate the score function at time step $t_k$. Assume that the score loss function $\mathcal{L}_k(\theta)$ satisfies the Assumptions 2 and 4, for all $k \in [0, K]$, and the learning rate for estimating $\mathcal{L}_k$ using SGD satisfies $0 \leq \eta \leq \frac{1}{\kappa}$, then with probability at least $1 - \delta$*

$$\mathcal{E}_k^{\text{opt}} \leq \mathcal{O}\left(W^D \cdot d \cdot \sqrt{\frac{\log\left(\frac{2}{\delta}\right)}{n_k}}\right). \tag{34}$$

**Proof Outline**

We study mini-batch SGD on the population loss $\mathcal{L}_k$

(i) $L$-smoothness, (ii) the (PL) condition with constant $\mu$, and (iii) unbiased mini-batch gradients with variance scaling as $\frac{\sigma^2}{b_i}$ . The algorithm is

$$\theta_{i+1} = \theta_i - \eta g_i, \qquad g_i = \frac{1}{b_i} \sum_{j=1}^{b_i} \nabla \ell(\theta_i; z_{i,j}) \tag{35}$$

with constant step size $\eta \leq 1/L$ and increasing batch size $b_i = \lceil \beta i \rceil$.

The proof starts from the smoothness (descent) inequality applied to the stochastic update:

$$\mathcal{L}_k(\theta_{i+1}) \leq \mathcal{L}_k(\theta_i) - \eta\langle\nabla\mathcal{L}_k(\theta_i), g_i\rangle + \frac{L\eta^2}{2}\|g_i\|_2^2. \tag{36}$$

Taking conditional expectation given $\theta_i$, unbiasedness yields $[\langle\nabla\mathcal{L}_k(\theta_i), g_i\rangle \mid \theta_i] = \|\nabla\mathcal{L}_k(\theta_i)\|_2^2$, and the mini-batch variance bound implies $[\|g_i\|_2^2 \mid \theta_i] \leq \|\nabla\mathcal{L}_k(\theta_i)\|_2^2 + \sigma^2/b_i$. Using $\eta \leq 1/L$ to simplify constants, we obtain the one-step recursion

$$[\mathcal{L}_k(\theta_{i+1}) - \mathcal{L}_k^\star] \leq (1 - \eta\mu)\,[\mathcal{L}_k(\theta_i) - \mathcal{L}_k^\star] + \frac{L\eta^2\sigma^2}{2b_i}, \tag{37}$$

where the PL condition converts $\|\nabla\mathcal{L}_k(\theta_i)\|_2^2$ into a multiple of $\mathcal{L}_k(\theta_i) - \mathcal{L}_k^\star$. Unrolling this recursion yields a geometric "optimization" term plus a "noise" term:

$$[\mathcal{L}_k(\theta_T) - \mathcal{L}_k^\star] \leq (1 - \eta\mu)^{T-1}\Delta_1 + \sum_{t=1}^{T-1}(1 - \eta\mu)^{T-1-t}\frac{L\eta^2\sigma^2}{2b_t}, \quad \Delta_1 := [\mathcal{L}_k(\theta_1) - \mathcal{L}_k^\star]. \tag{38}$$

The increasing batch schedule $b_t = \lceil \beta t \rceil$ makes the variance term decay like $1/t$; the geometric weights ensure the sum is bounded by a constant multiple of $1/T$, giving

$$[\mathcal{L}_k(\theta_T) - \mathcal{L}_k^\star] \leq (1 - \eta\mu)^{T-1}\Delta_1 + \mathcal{O}\left(\frac{1}{T}\right). \tag{39}$$

Finally, if $n = \sum_{i=1}^T b_i$ denotes the total number of samples, then $b_i \asymp i$ implies $n = \Theta(T^2)$ and hence $1/T = \Theta(1/\sqrt{n})$. Substituting this relation yields the stated sample-based convergence rate

$$[\mathcal{L}_k(\theta_T) - \mathcal{L}_k^\star] \quad \leq (1 - \eta\mu)^{T-1}\Delta_1 + \mathcal{O}\left(\frac{1}{\sqrt{n}}\right) = \mathcal{O}\left(\frac{1}{\sqrt{n}}\right) \tag{40}$$

and in the large-$n$ regime the geometric transient is negligible, giving $[\mathcal{L}_k(\theta_T) - \mathcal{L}_k^\star] = \mathcal{O}(n^{-1/2})$. We use the quadratic growth inequality and the result of Lemma 3 to get the final result. The details of the analysis are given in Appendix C.2.

Combining the decomposition in Eq. (20) along with Lemmas 1–3, we obtain the following bound on $A(k)$ (defined in Eq. (18)) with probability at least $1 - \delta$

$$A(k) \leq \mathcal{O}\left(W^D{\cdot}d{\cdot}\sqrt{\frac{\log\left(\frac{2}{\delta}\right)}{n_k}}\right) + \mathcal{O}\left(W^D{\cdot}d{\cdot}\sqrt{\frac{\log\left(\frac{2}{\delta}\right)}{n_k}}\right) + \epsilon_{approx} \tag{41}$$

$$\leq \mathcal{O}\left(W^D{\cdot}d{\cdot}\sqrt{\frac{\log\left(\frac{2}{\delta}\right)}{n_k}}\right) + \epsilon_{approx}, \tag{42}$$

where in the second inequality we combine the first two terms appropriately. Setting the sample size

$$n_k = \Omega\left(W^{2D}{\cdot}d^2{\cdot}\log^2\left(\frac{4K}{\delta}\right)\left(\frac{\epsilon^{-4}}{\sigma_k^{-4}}\right)\right), \tag{43}$$

we ensure that $A(k) \leq \frac{\epsilon^2}{\sigma_k^2} = \frac{\epsilon^2}{1 - e^{-2(T - t_k)}}$ for all $k \in \{0, \ldots, K\}$. Summing over all time steps, we obtain with probability at least $1 - \delta$

$$\sum_{k=0}^{K} A(k)(t_{k+1} - t_k) \leq \sum_{k=0}^{K} \frac{\epsilon^2}{1 - e^{-2(T-t_k)}}(t_{k+1} - t_k) \tag{44}$$

$$\leq \int_0^{T-\kappa} \frac{\epsilon^2}{1 - e^{-2(T-t)}} dt \leq \epsilon^2 \left( T + \log \frac{1}{\kappa} \right). \tag{45}$$

Note that the term $\left( \log^2 \left( \frac{4K}{\delta} \right) \right)$ appears in the upper bound for $n_k$ in Eq. (43) since we have to take a union bound for Lemma 2 and Lemma 3 and then take a union bound over $K$ discretization steps. Substituting this bound into Eq. (18), and then substituting the result into Eq. (16), we obtain that with probability at least $1 - \delta$.

$$\mathrm{TV}(p_{t_0}, \hat{p}_{t_0}) \leq \mathcal{O}(\exp^{-T}) + \mathcal{O}\left( \frac{1}{\sqrt{K}} \right) + \mathcal{O}\left( \epsilon \cdot \sqrt{\left( T + \log \frac{1}{\kappa} \right)} \right) + \epsilon_{approx} \tag{46}$$

Finally, by choosing $T = \Omega\left( \log\left( \frac{1}{\epsilon} \right) \right)$, $\kappa = \Omega(\epsilon)$ and $K = \Omega(\epsilon^{-2})$, we conclude that with probability at least $1 - \delta$

$$\mathrm{TV}(p_{t_0}, \hat{p}_{t_0}) \leq \mathcal{O}(\epsilon) + \epsilon_{approx}, \tag{47}$$

completing the proof of Theorem 1. $\qquad\square$

In summary, our work provides a principled decomposition of the errors in score-based generative models, highlighting how each component contributes to the overall sample complexity. This leads to the first finite sample complexity bound of $\widetilde{\mathcal{O}}(\epsilon^{-4})$ for diffusion models without assuming access to the empirical minimizer of the score estimation function.

## 4 Conclusion and Future Work

In this work, we investigate the sample complexity of training diffusion models via score estimation using neural networks. We derive a sample complexity bound of $\widetilde{\mathcal{O}}(\epsilon^{-4})$, which, to our knowledge, is the first such result that does not assume access to an empirical risk minimizer of the score estimation loss. Notably, our bound does not depend exponentially on the number of neural network parameters. For comparison, the best-known existing result achieves a bound of $\widetilde{\mathcal{O}}(\epsilon^{-5})$, but it crucially assumes access to an ERM. All prior results establishing sample complexity bounds for diffusion models have made this assumption. Our contribution is the first to establish a sample complexity bound for diffusion models under the more realistic setting where exact access to empirical risk minimizer of the score estimation loss is not available.

We note that although this work focuses on continuous data distributions, subsequent work has established a $\widetilde{\mathcal{O}}(\epsilon^{-2})$ sample complexity for discrete-state diffusion models in (Srikanth et al., 2026). Whether an analogous $\widetilde{\mathcal{O}}(\epsilon^{-2})$ sample complexity is achievable for continuous diffusion models remains open, although such a result has recently been established for rectified flow matching in (Sahoo et al., 2026).

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

## A  Note about (Gupta et al., 2024)

We note that the comparison in this paper uses the results from the updated arXiv version of their work (in Nov 2025), which incorporates a correction to an error identified in the earlier version of our work (available at arXiv).

## B  Score Estimation Algorithm

In this section, we provide a detailed description of the algorithm used for estimating the score function in diffusion models.

---

**Algorithm 1** Denoising Diffusion Probabilistic Model (DDPM)

---

1: **Input:** Dataset $\mathcal{D}$, timesteps $T$, stop time $t_{\text{stop}}$, schedule $\{\beta_t\}_{t=1}^T$, network $\epsilon_\theta$, learning rate $\eta$, iterations $K$
2: Precompute: $\alpha_t = 1 - \beta_t$, $\bar{\alpha}_t = \prod_{s=1}^t \alpha_s$
    **Training (Score Estimation)**
3: **for** $i = 1$ to $N$ **do**
4:     Sample $x_j \sim \mathcal{D}$, $k \sim \text{Uniform}([1, T])$, $\epsilon_k \sim \mathcal{N}(0, I)$ for $i = 1, \ldots, n$
5:     $x_{t_i} = e^{-t_k} x_i + \sqrt{1 - e^{-2t_k}} \epsilon_i$
6:     Compute loss: $\hat{L}(\theta) = \|\epsilon_i - \epsilon_\theta(x_{t_i}, t_i)\|^2$
7:     Update $\theta \leftarrow \theta - \eta_k \cdot \nabla_\theta \hat{L}(\theta)$
8: **end for**
    **Sampling**
9: Sample $x_T \sim \mathcal{N}(0, I)$
10: **for** $t = T$ down to $t_{\text{stop}} + 1$ **do**
11:     $z \sim \mathcal{N}(0, I)$ if $t > 1$ else $z = 0$
12:     $\hat{\epsilon} = \epsilon_\theta(x_t, t)$
13:     $\tilde{\mu}_t = \frac{1}{\sqrt{\alpha_t}} \left( x_t - \frac{\beta_t}{\sqrt{1 - \bar{\alpha}_t}} \hat{\epsilon} \right)$
14:     $x_{t-1} = \tilde{\mu}_t + \sqrt{\beta_t} \cdot z$
15: **end for**
16: **Return** $x_{t_{\text{stop}}}$

---

## C  Proofs of Intermediate Lemmas

In this section, we present the proofs of intermediate lemmas used to bound the statistical error and optimization error in our analysis.

### C.1  Bounding the Statistical Error

*Proof.* Let us define the population loss at time $t_k$ for $k \in [0, K]$ as

$$\mathcal{L}_k(\theta) = \mathbb{E}_{x \sim p_{t_k}} \|s_\theta(x, t_k) - \nabla \log p_{t_k}(x)\|^2, \tag{48}$$

where $s_\theta$ denotes the score function estimated by a neural network parameterized by $\theta$, and $x$ denotes samples at time $t$ used in Algorithm 1. The corresponding empirical loss is defined as:

$$\widehat{\mathcal{L}}_k(\theta) = \frac{1}{n} \sum_{i=1}^n \|s_\theta(x_i, t_k) - \nabla \log p_{t_k}(x_i)\|^2. \tag{49}$$

Let $\theta_k^a$ and $\theta_k^b$ be the minimizers of $\mathcal{L}_k(\theta)$ and $\widehat{\mathcal{L}}_t(\theta)$, respectively, corresponding to score functions $s_{t_k}^a$ and $s_{t_k}^b$. By the definitions of minimizers, we can write

$$\mathcal{L}_k(\theta_k^b) - \mathcal{L}_k(\theta_k^a) \leq \mathcal{L}_k(\theta_k^b) - \mathcal{L}_k(\theta_k^a) + \widehat{\mathcal{L}}_k(\theta_k^a) - \widehat{\mathcal{L}}_k(\theta_k^b) \tag{50}$$

$$\leq \underbrace{\left| \mathcal{L}_k(\theta_k^b) - \widehat{\mathcal{L}}_t(\theta_k^b) \right|}_{(I)} + \underbrace{\left| \mathcal{L}_k(\theta_k^a) - \widehat{\mathcal{L}}_t(\theta_k^a) \right|}_{(II)}. \tag{51}$$

Note that the right-hand side of (50) is greater than the left-handeft-hand side since we have added the quantity $\widehat{\mathcal{L}}_t(\theta_k^a) - \widehat{\mathcal{L}}_t(\theta_k^b)$ which is strictly positive since $\theta_K^b$ is the minimizer of the function $\widehat{\mathcal{L}}_k(\theta)$ by definition. We then take the absolute value on both sides of the (51) to get

We now bound terms (I) and (II) using generalization results. From Lemma 5 (Theorem 26.5 of (Shalev-Shwartz & Ben-David, 2014)), if the loss function $\widehat{\mathcal{L}}(\theta)$ is uniformly bounded over the parameter space $\Theta'' = \{\theta_k^a, \theta_k^b\}$, then with probability at least $1 - \delta$, we have

$$\left| \mathcal{L}_k(\theta) - \widehat{\mathcal{L}}_t(\theta) \right| \leq \widehat{R}(\Theta'') + \mathcal{O}\left( \sqrt{\frac{\log \frac{1}{\delta}}{n}} \right), \quad \forall \, \theta \in \Theta'' \tag{52}$$

where $\widehat{R}(\Theta'')$ denotes the empirical Rademacher complexity of the function class restricted to $\Theta''$. Now since $x$ is not bounded, this result does not hold. We then define the following two functions

$$\mathcal{L}'_k(\theta) = \mathbb{E}_{x \sim \mu_{t_k}} \left\| v_\theta(x, t_k) - v_{t_k}(x) \right\|^2, \tag{53}$$

and

$$\widehat{\mathcal{L}}'_k(\theta) = \frac{1}{n} \sum_{i=1}^{n} \left\| v_\theta(x_i, t_k) - v_{t_k}(x_i) \right\|^2. \tag{54}$$

where we define the functions

$$(v_t(x))_j = \begin{cases} (\nabla \log p_t(x))_j & \text{if } |\frac{x - e^{-t}x_0}{\sigma_t^2}|_j \leq \kappa \\ 0 & \text{if } |\frac{x - e^{-t}x_0}{\sigma_t^2}|_j \geq \kappa \end{cases} \tag{55}$$

and

$$(v_\theta(x, t))_j = \begin{cases} (s_\theta(x, t))_j & \text{if } |\frac{x - e^{-t}x_0}{\sigma_t^2}|_j \leq \kappa \\ 0 & \text{if } |\frac{x - e^{-t}x_0}{\sigma_t^2}|_j \geq \kappa \end{cases} \tag{56}$$

Here $(v_t(x))_j, (\nabla \log p_t(x))_j$, $(v_\theta(x, t))_j$ and $(s_\theta(x, t))_j$ denote the $j^{th}$ co-ordinate of $v_t(x)$, $(\nabla \log p_t(x))$, $v_\theta(x, t)$ and $s_\theta(x, t)$ respectively. Further, $|\frac{x - e^{-t}x_0}{\sigma_t^2}|_j$ denotes the $j^{th}$ co-ordinate of the $i^{th}$ sample of the score function in $\widehat{\mathcal{L}}_k(\theta)$ which is given by $\log p_t(x) = |\frac{x - e^{-t}x_0}{\sigma_t^2}|$.

Note that the functions $v_t(x)$ and $v_\theta(x, t)$ are uniformly bounded. Thus using Theorem 26.5 of (Shalev-Shwartz & Ben-David, 2014) we have with probability at least $1 - \delta$,

$$\left| \mathcal{L}'_k(\theta) - \widehat{\mathcal{L}}'_k(\theta) \right| \leq \widehat{R}(\theta) + \mathcal{O}\left( \sqrt{\frac{\log \frac{1}{\delta}}{n}} \right), \quad \forall \, \theta \in \Theta''. \tag{57}$$

Since $\Theta'' = \{\theta_a, \theta_b\}$ is a finite class (just two functions). We can apply Lemma E.5 to bound the empirical Rademacher complexity $\widehat{R}(\theta)$ in terms of the Rademacher complexity $R(\theta)$ of the function class $\Theta''$. Since $\widehat{R}(\theta) = \frac{1}{m}\mathbb{E}_\sigma \left[ \max_{\theta \in \Theta''} \sum_{i=1}^{n} f(\theta)\sigma_i \right]$, applying Lemma E.5, we have with probability at least $1 - 2\delta$

$$\left| \mathcal{L}'_k(\theta) - \widehat{\mathcal{L}}'_k(\theta) \right| \leq \mathcal{O}\left( \frac{d \cdot W^D}{n} \right) + \mathcal{O}\left( \sqrt{\frac{\log \frac{1}{\delta}}{n}} \right), \quad \forall \, \theta \in \Theta''. \tag{58}$$

This yields that with probability at least $1 - \delta$ we have

$$\left| \mathcal{L}'_k(\theta) - \widehat{\mathcal{L}}'_k(\theta) \right| \leq \mathcal{O}\left( d \cdot W^D \cdot \sqrt{\frac{\log \frac{1}{\delta}}{n}} \right), \qquad \forall \, \theta \in \Theta'' \tag{59}$$

From this we have

$$\left[ \left| \mathcal{L}'_k(\theta) - \widehat{\mathcal{L}}_k(\theta) \right| \right] \leq \mathcal{O}\left( d \cdot \sqrt{\frac{\log \frac{1}{\delta}}{n}} \right) \tag{60}$$

Now consider the probability of the event

$$A_{i,j} = \left\{ \left| \frac{x_i - e^{-t}(x_0)_i}{\sigma_t^2} \right|_j \geq \kappa \right\} \tag{61}$$

Where $\left| \frac{x_i - e^{-t}(x_0)_i}{\sigma_t^2} \right|_j$ denotes the $k^{th}$ co-ordinate of of the $i^{th}$ sample of the score function given by $\left| \frac{x_i - e^{-t}(x_0)_i}{\sigma_t^2} \right|$ We have the probability of this event upper bounded as

$$P\left( \left| \frac{x_i - e^{-t}x_0}{\sigma_t^2} \right|_j \geq \kappa \right) = \mathbb{E}_z P\left( \left| \frac{x_i - e^{-t}x_0}{\sigma_t^2} \right|_j \geq \kappa \bigg| x_0 \right) \tag{62}$$

$$\leq \exp\left( -\kappa^2(1 - e^{-t}) \right) \tag{63}$$

$$\leq \exp\left( -\kappa^2 \right) \tag{64}$$

We get (63) from (62) since the score variable is conditionally normal given $x_0$.

Setting $\kappa = \log\left( \frac{dn}{\delta} \right)$, we have

$$P\left( \left| \frac{x_i - e^{-t}x_0}{\sigma_t^2} \right|_j \geq \kappa \right) \leq \frac{\delta}{dn} \tag{65}$$

If we denote the event $A = \{\widehat{L}'(\theta) = \widehat{L}(\theta)\}$, then by union bound we have $P(A) = P(\cup_{i,j} A_{i,j}) \leq \sum_{i,j} P(A_{i,j}) \leq \delta$. Let event $B$ denote the failure of the generalization bound, i.e.,

$$B := \left\{ \left| \mathcal{L}'_t(\theta) - \widehat{\mathcal{L}}'_t(\theta) \right| > \widehat{R}(\Theta'') + \mathcal{O}\left( \sqrt{\frac{\log \frac{1}{\delta}}{n}} \right) \right\}. \tag{66}$$

From above, we know $\mathbb{P}(B) \leq \delta$ under the boundedness condition. Therefore, by the union bound, we have

$$\mathbb{P}(A \cup B) \leq \mathbb{P}(A) + \mathbb{P}(B) \leq 2\delta, \tag{67}$$

$$\implies \mathbb{P}(A^c \cap B^c) = 1 - P(A \cup B) \geq 1 - 2\delta. \tag{68}$$

On this event $(A^c \cap B^c)$, we have $\widehat{\mathcal{L}'}(\theta) = \widehat{\mathcal{L}}(\theta)$. Hence, with probability at least $1 - 2\delta$, we have

$$|\mathcal{L}_k(\theta_k^b) - \mathcal{L}_k(\theta_k^a)| \le \left|\mathcal{L}_k(\theta_k^b) - \widehat{\mathcal{L}}_t(\theta_k^b)\right| + \left|\mathcal{L}_k(\theta_k^a) - \widehat{\mathcal{L}}_t(\theta_k^a)\right|. \tag{69}$$

$$\le \left|\mathcal{L}_k(\theta_k^b) - \widehat{\mathcal{L}'}_t(\theta_k^b)\right| + \left|\mathcal{L}_k(\theta_k^a) - \widehat{\mathcal{L}'}_t(\theta_k^a)\right|. \tag{70}$$

$$= \left|\mathcal{L}_k(\theta_k^b) - \mathcal{L}'_t(\theta_k^b)\right| + \left|\mathcal{L}_k(\theta_k^a) - \mathcal{L}'_t(\theta_k^a)\right|.$$
$$+ \left|\mathcal{L}'_t(\theta_k^b) - \widehat{\mathcal{L}'}_t(\theta_k^b)\right| + \left|\mathcal{L}_k(\theta_k^a) - \widehat{\mathcal{L}'}_t(\theta_k^a)\right|. \tag{71}$$

$$\le |\mathcal{L}'_t(\theta_k^a) - \mathcal{L}_t(\theta_k^a)| + |\mathcal{L}'_t(\theta_k^b) - \mathcal{L}_t(\theta_k^b)| + \mathcal{O}\left(d \cdot W^D \cdot \sqrt{\frac{\log\frac{1}{\delta}}{n}}\right) \tag{72}$$

In order to bound $|\mathcal{L}_k(\theta) - \mathcal{L}'_t(\theta)|$ we have the following

$$|\mathcal{L}_k(\theta) - \mathcal{L}'_t(\theta)| \le \sum_{j=1}^{d} \mathbb{E}_{x_j \sim (u_{t_k})_j} |(s_\theta(x, t_k)) - (\nabla \log p_{t_k}(x))|_j^2 - \mathbb{E}_{x \sim u_t} |(v_t(x))_k - (v_\theta(x, t))|_j^2 \tag{73}$$

$$\le \sum_{j=1}^{d} \mathbb{E}_{x_j \sim (u_{t_k})_j} \left( |(\nabla \log p_{t_k}) - (s_\theta(x, t_k))|_j^2 \mathbf{1}_{\left|\frac{x - e^{-t}(x_0)_i}{\sigma_t^2}\right|_j \ge \kappa} \right) \tag{74}$$

$$\le \sum_{j=1}^{d} \mathbb{E}_{x_j \sim (u_{t_k})} \left( \left|\frac{x - e^{-t}(x_0)}{\sigma_t^2} - (s_\theta(x, t))_k\right|_j^2 \mathbf{1}_{\left|\frac{x - e^{-t}(x_0)}{\sigma_t^2}\right|_j \ge \kappa} \right) \tag{75}$$

$$\le 2 \sum_{j=1}^{d} \mathbb{E}_{x_j \sim (u_{t_k})_j} \left( \left|\frac{x - e^{-t}(x_0)}{\sigma_t^2}\right|_j^2 \mathbf{1}_{\left|\frac{x - e^{-t}(x_0)}{\sigma_t^2}\right|_j \ge \kappa} \right)$$

$$+ \sum_{j=1}^{d} 2\mathbb{E}_{x_j \sim (u_{t_k})_j} \left( (s_\theta(x, t))_j^2 \mathbf{1}_{\left|\frac{x - e^{-t}(x_0)}{\sigma_t^2}\right|_j \ge \kappa} \right) \tag{76}$$

$$\le 2 \sum_{j=1}^{d} \mathbb{E}_{x_j \sim (u_{t_k})_j} \left( \left|\frac{x - e^{-t}(x_0)}{\sigma_t^2}\right|_j^2 \mathbf{1}_{\left|\frac{x - e^{-t}(x_0)}{\sigma_t^2}\right|_j \ge \kappa} \right)$$

$$+ \sum_{j=1}^{d} C_{\Phi''} \mathbb{E}_{x_j \sim (u_{t_k})_j} \left( \left|\frac{x - e^{-t}(x_0)}{\sigma_t^2}\right|_j \mathbf{1}_{\left|\frac{x - e^{-t}(x_0)}{\sigma_t^2}\right| \ge \kappa} \right) \tag{77}$$

$$\leq \left(\frac{4 + 2C_{\Phi''}}{\sigma_t{}^2}\right) \sum_{k=1}^{d} \underbrace{\mathbb{E}_{x_j \sim (u_{t_k})_j} \left( |x|_j^2 \mathbf{1}_{\left|\frac{x - e^{-t}(x_0)}{\sigma_t^2}\right|_j \geq \kappa} \right)}_{I}$$

$$+ \frac{2}{\sigma_t{}^2} \underbrace{\mathbb{E}_{x_j \sim (u_{t_k})_j} \left( |x_0|_j^2 \mathbf{1}_{\left|\frac{x - e^{-t}(x_0)}{\sigma_t^2}\right|_j \geq \kappa} \right)}_{II} \tag{78}$$

We get Equation (76) from Equation (75) by using the identity $(a - b)^2 \leq 2|a|^2 + 2|b|^2$. We get Equation (77) from Equation (76) by using Lemma 8. We get Equation (78) from Equation (77) by using the identity $(a - b)^2 \leq 2|a|^2 + 2|b|^2$ again. Now we separately obtain upper bounds for the terms $I$ and $II$ as follows.

$$\sum_{j=1}^{d} \mathbb{E}_{x_j \sim (u_{t_k})_j} \left( |x_j|^2 \mathbf{1}_{\left|\frac{x_i - e^{-t}(x_0)_i}{\sigma_t^2}\right|_j \geq \kappa} \right) \tag{79}$$

$$= \sum_{j=1}^{d} \mathbb{E}_{x_j \sim (u_{t_k})_j} \left( |x_j|^2 \mathbf{1}_{\left|\frac{x_i - e^{-t}(x_0)_i}{\sigma_t^2}\right|_j \geq \kappa} \right) \tag{80}$$

$$\leq \sum_{j=1}^{d} \mathbb{E}_{x_0} \mathbb{E}_{x_j \sim (u_{t_k})_j | x_0} \left( |x_j|^2 \mathbf{1}_{\left|\frac{x_i - e^{-t}(x_0)_i}{\sigma_t^2}\right|_j \geq \kappa} \right) \tag{81}$$

$$\leq \sum_{j=1}^{d} \mathbb{E}_{x_0} \mathbb{E}_{x_j \sim (u_t)_k | x_0} \left( |x_j|^2 \mathbf{1}_{\left|\frac{x_i - e^{-t}(x_0)_i}{\sigma_t^2}\right|_j \geq \kappa} \right) P\left( \left|\frac{x_i - e^{-t}(x_0)_i}{\sigma_t^2}\right| \geq \kappa \Big| x_0 \right) \tag{82}$$

$$\leq \exp\left(-\kappa^2\right) \sum_{j=1}^{d} \mathbb{E}_{x_0} \mathbb{E}_{x_0 \sim (u_t)_j | x_0} \left( |x_k|^2 \Big|_j \mathbf{1}_{\left|\frac{x_i - e^{-t}(x_0)_i}{\sigma_t^2}\right|_j \geq \kappa} \right) \tag{83}$$

$$\leq \exp\left(-\kappa^2\right) \sum_{k=1}^{d} \mathbb{E}_{x_0} \left( \sigma_t^2 + \sigma_t^2 . \kappa . \sigma_t^2 . \frac{\phi(\kappa \cdot \sigma_t^2)}{1 - \Phi((\kappa \cdot \sigma_t^2))} \right) \tag{84}$$

$$\leq \exp\left(-\kappa^2\right) \sum_{k=1}^{d} \mathbb{E}_z \left( 2 . \sigma_t^2 \right) \tag{85}$$

$$\leq \mathcal{O}\left(\exp\left(-\kappa^2\right)\right) \tag{86}$$

We get Equation (84) from Equation (83) by using Lemma 7. We get Equation (86) from Equation (85) from Assumption 1 and by using the upper bound on the Mill's ration which implies that $\frac{\phi(\kappa)}{1 - \Phi(\kappa)} \leq \kappa + \frac{1}{\kappa}$.

We get Equation (86) from Equation (85) from Assumption 1, which implies that the second moment of $z$ is bounded.

$$\sum_{j=1}^{d} \mathbb{E}_{x_j \sim (u_{t_k})_j} \left( |x|_0^2 \mathbf{1}_{\left| \frac{x_i - e^{-t}(x_0)_i}{\sigma_t^2} \right|_j \geq \kappa} \right) \tag{87}$$

$$= \sum_{j=1}^{d} \mathbb{E}_{x_k \sim (u_t)_k} \left( |x_0|^2 \mathbf{1}_{\left| \frac{x_i - e^{-t}(x_0)_i}{\sigma_t^2} \right|_j \geq \kappa} \right) \tag{88}$$

$$\leq \sum_{j=1}^{d} \mathbb{E}_{x_0} \mathbb{E}_{x_k \sim (u_t)_k | x_0} \left( |x_0|^2 \mathbf{1}_{\left| \frac{x_i - e^{-t}(x_0)_i}{\sigma_t^2} \right|_j \geq \kappa} \right) \tag{89}$$

$$\leq \sum_{j=1}^{d} \mathbb{E}_{x_0} \mathbb{E}_{x_k \sim (u_t)_k | x_0} \left( |x_0|^2 \mathbf{1}_{\left| \frac{x_i - e^{-t}(x_0)_i}{\sigma_t^2} \right|_j \geq \kappa} \right) P\left( \left| \frac{x_k - t z_k}{1 - t} \right|_j \geq \kappa | x_0 \right) \tag{90}$$

$$\leq \exp\left(-\kappa^2\right) \sum_{j=1}^{d} \mathbb{E}_{x_0} |x_0|^2 \tag{91}$$

$$\leq \mathcal{O}\left(\exp\left(-\kappa^2\right)\right) \tag{92}$$

Setting $\kappa = \log \frac{dn}{\delta}$ Plugging Equation (86), (92) into Equation (78). Then we have

$$|\mathcal{L}_k(\theta) - \mathcal{L}'_t(\theta)| \leq \mathcal{O}\left(\exp\left(-\kappa^2\right)\right), \quad \forall \theta = \{\theta_k^a, \theta_k^b\} \tag{93}$$

$$\leq \mathcal{O}\left(\frac{\delta}{dn}\right) \tag{94}$$

Now plugging Equation (93) into Equation (72) we get with probability at least $1 - 2\delta$

$$|\mathcal{L}_k(\theta_k^b) - \mathcal{L}_k(\theta_k^a)| \leq \mathcal{O}\left( d \cdot W^D \cdot \sqrt{\frac{\log \frac{1}{\delta}}{n}} \right) \tag{95}$$

Finally, using the Polyak-Łojasiewicz (PL) condition for $\mathcal{L}_k(\theta)$, from Assumption 2, we have from the quadratic growth condition of PL functions the following,

$$\|\theta_k^a - \theta_k^b\|^2 \leq \mu \left| \mathcal{L}_k(\theta_k^a) - \mathcal{L}_k(\theta_k^b) \right|, \tag{96}$$

and applying Lipschitz continuity of the velocity fields with respect to parameter $x$

$$\|v^{\theta_k^a}(x, t_k) - v^{\theta_k^b}(x, t_k)\|^2 \leq L_t \cdot \|\theta_k^a - \theta_k^b\|^2 \tag{97}$$

$$\leq L_t \cdot \mu \left|\mathcal{L}_k(\theta_k^a) - \mathcal{L}_k(\theta_k^b)\right| \tag{98}$$

$$\leq \mathcal{O}\left(d \cdot W^D \cdot \sqrt{\frac{\log \frac{1}{\delta}}{n}}\right). \tag{99}$$

Here $L_t$ is the Lipschitz parameter of the neural networks. It is always possible to obtain this Lipschitz constant as the quantity $\|v^a(x, t) - v^b(x, t)\|^2 \leq L_t$ is non-zero only over a finite domain of $x$. Taking expectation with respect to $x$, we obtain the following.

$$\mathbb{E}_{x \sim u_{t_k}}\|s^{\theta_k^a}(x, t_k) - s^{\theta_k^b}(x, t_k)\|^2 \leq 2\mathbb{E}_{x \sim u_t}\|s^{\theta_k^a}(x, t_k) - s^{\theta_k^b}(x, t_k) - v_t^a(x) - v_t^b(x)\|^2$$

$$+ 2\mathbb{E}_{x \sim u_t}\|v^{\theta_k^a}(x, t_k) - v^{\theta_k^b}(x, t_k)\|^2 \tag{100}$$

$$\leq 4\mathbb{E}_{x \sim u_t}\|v^{\theta_k^a}(x, t_k) - s^{\theta_k^a}(x, t_k)\|^2 \tag{101}$$

$$+ 4\mathbb{E}_{x \sim u_t}\|s^{\theta_k^b}(x, t_k) - v^{\theta_k^b}(x, t_k)\|^2$$

$$+ 4\mathbb{E}_{x \sim u_t}\|v_t^a(x) - v_t^b(x)\|^2 \tag{102}$$

$$\leq \mathcal{O}(\kappa^{-2}) + \mathcal{O}\left(d \cdot W^D \cdot \sqrt{\frac{\log \frac{1}{\delta}}{n}}\right). \tag{103}$$

$$\leq \mathcal{O}(\frac{\delta}{dn}) + \mathcal{O}\left(d \cdot W^D \cdot \sqrt{\frac{\log \frac{1}{\delta}}{n}}\right). \tag{104}$$

$$\leq \mathcal{O}\left(d \cdot W^D \cdot \sqrt{\frac{\log \frac{1}{\delta}}{n}}\right). \tag{105}$$

This completes the proof. Note that the quantities $4\mathbb{E}_{x \sim u_t}\|u_t^a(x) - v_t^a(x)\|^2$ and $4\mathbb{E}_{x \sim u_t}\|u_t^a(x) - v_t^a(x)\|^2$ are bounded in the same manner as is done in Equation (92).

$\square$

## C.2 Bounding Optimization Error

The optimization error ($\mathcal{E}_{\text{opt}}$) accounts for the fact that gradient-based optimization does not necessarily find the optimal parameters due to limited steps, local minima, or suboptimal learning rates. This can be bounded as follows.

*Proof.* By $L$-smoothness with $\theta_{i+1} = \theta_i - \eta g_i$,

$$\mathcal{L}(\theta_{i+1}) \leq \mathcal{L}(\theta_i) - \eta\langle\nabla\mathcal{L}(\theta_i), g_i\rangle + \frac{L\eta^2}{2}\|g_i\|_2^2.$$

Taking conditional expectation given $\theta_i$ and using unbiasedness,

$$\mathbb{E}[\langle\nabla\mathcal{L}(\theta_i), g_i\rangle \mid \theta_i] = \|\nabla\mathcal{L}(\theta_i)\|_2^2.$$

Moreover,

$$\mathbb{E}\left[\|g_i\|_2^2 \mid \theta_i\right] = \mathbb{E}\left[\|\nabla\mathcal{L}(\theta_i) + (g_i - \nabla\mathcal{L}(\theta_i))\|_2^2 \mid \theta_i\right]$$

$$= \|\nabla\mathcal{L}(\theta_i)\|_2^2 + \mathbb{E}\left[\|g_i - \nabla\mathcal{L}(\theta_i)\|_2^2 \mid \theta_i\right]$$

$$\leq \|\nabla\mathcal{L}(\theta_i)\|_2^2 + \frac{\sigma^2}{b_i}.$$

Combining the last three displays,

$$\mathbb{E}[\mathcal{L}(\theta_{i+1}) \mid \theta_i] \le \mathcal{L}(\theta_i) - \eta \|\nabla\mathcal{L}(\theta_i)\|_2^2 + \frac{L\eta^2}{2}\left(\|\nabla\mathcal{L}(\theta_i)\|_2^2 + \frac{\sigma^2}{b_i}\right).$$

Rearranging and using $\eta \le 1/L$ (so that $\eta - \frac{L\eta^2}{2} \ge \eta/2$),

$$\mathbb{E}[\mathcal{L}(\theta_{i+1}) \mid \theta_i] \le \mathcal{L}(\theta_i) - \frac{\eta}{2}\|\nabla\mathcal{L}(\theta_i)\|_2^2 + \frac{L\eta^2\sigma^2}{2b_i}.$$

Subtracting $\mathcal{L}^\star$ and applying the PL inequality $\|\nabla\mathcal{L}(\theta_i)\|_2^2 \ge 2\mu(\mathcal{L}(\theta_i) - \mathcal{L}^\star)$ yields

$$\mathbb{E}[\mathcal{L}(\theta_{i+1}) - \mathcal{L}^\star \mid \theta_i] \le (1 - \eta\mu)(\mathcal{L}(\theta_i) - \mathcal{L}^\star) + \frac{L\eta^2\sigma^2}{2b_i}.$$

Taking total expectation gives the recursion

$$\Delta_{i+1} \le (1 - \eta\mu)\Delta_i + \frac{L\eta^2\sigma^2}{2b_i}. \tag{106}$$

Let $\rho := 1 - \eta\mu \in (0,1)$ and choose the increasing batch size schedule

$$b_i = \lceil \beta i \rceil \qquad \text{for some constant } \beta > 0.$$

Unrolling (106) for $i = 1, \ldots, T-1$ gives

$$\Delta_T \le \rho^{T-1}\Delta_1 + \sum_{t=1}^{T-1} \rho^{T-1-t}\frac{L\eta^2\sigma^2}{2b_t}. \tag{107}$$

Since $b_t \ge \beta t$, we bound

$$\sum_{t=1}^{T-1} \rho^{T-1-t}\frac{1}{b_t} \le \frac{1}{\beta}\sum_{t=1}^{T-1}\frac{\rho^{T-1-t}}{t}.$$

Let $k := T - 1 - t$, so $t = T - 1 - k$ and $k = 0, 1, \ldots, T-2$. Then

$$\sum_{t=1}^{T-1}\frac{\rho^{T-1-t}}{t} = \sum_{k=0}^{T-2}\frac{\rho^k}{T-1-k} \le \frac{1}{T-1}\sum_{k=0}^{\infty}\rho^k = \frac{1}{T-1}\cdot\frac{1}{1-\rho} = \frac{1}{T-1}\cdot\frac{1}{\eta\mu}.$$

Plugging this into (107) yields

$$\Delta_T \le \rho^{T-1}\Delta_1 + \frac{L\eta^2\sigma^2}{2}\cdot\frac{1}{\beta}\cdot\frac{1}{\eta\mu}\cdot\frac{1}{T-1} = \rho^{T-1}\Delta_1 + \frac{L\eta\sigma^2}{2\beta\mu}\cdot\frac{1}{T-1}. \tag{108}$$

Thus, the suboptimality decays as $\mathcal{O}(1/T)$ up to a transient geometric term:

$$\mathbb{E}[\mathcal{L}(\theta_T) - \mathcal{L}^\star] \le (1 - \eta\mu)^{T-1}\Delta_1 + \mathcal{O}\left(\frac{1}{T}\right).$$

Let $n$ denote the total number of samples used up to iteration $T$:

$$n := \sum_{i=1}^{T} b_i.$$

With $b_i = \lceil \beta i \rceil$, we have the crude lower bound

$$n \ge \sum_{i=1}^{T} \beta i = \frac{\beta T(T+1)}{2} \ge \frac{\beta T^2}{2},$$

hence

$$\frac{1}{T-1} \ \leq \ \frac{2}{T} \ \leq \ 2\sqrt{\frac{2}{\beta n}} \ = \ \mathcal{O}\left(\frac{1}{\sqrt{n}}\right).$$

Substituting this into (108) gives

$$\mathbb{E}\big[\mathcal{L}(\theta_T) - \mathcal{L}^\star\big] \ \leq \ (1-\eta\mu)^{T-1}\Delta_1 \ + \ \mathcal{O}\left(\frac{1}{\sqrt{n}}\right).$$

In particular, for sufficiently large $n$ (so the transient term is negligible),

$$\boxed{\mathbb{E}\big[\mathcal{L}(\theta_T) - \mathcal{L}^\star\big] \ = \ \mathcal{O}\left(\frac{1}{\sqrt{n}}\right).}$$

Note that $\hat{s}_{t_k}$ and $\hat{\theta}_k$ denote our estimate of the loss function and associated parameter obtained from the SGD. Also note that $\mathcal{L}^*$ is the loss function corresponding whose minimizer is the neural network $s_{t_k}^a$ and the neural parameter $\theta_k^a$ is our estimated score parameter. Thus applying the quadratic growth inequality.

$$\|\hat{s}_{t_k}(x,t_k) - s_{t_k}^a(x,t_k)\|^2 \leq L.\|\hat{\theta}_k - \theta_a^k\|^2 \leq \|[\mathcal{L}(\theta_k) - \mathcal{L}^*]\| \tag{109}$$

$$\leq \mathcal{O}\left(\frac{1}{\sqrt{n}}\right) \tag{110}$$

From lemma 2 we have with probability $1-\delta$ that

$$\|s_{t_k}^a(x,t_k) - s_{t_k}^b(x,t_k)\|^2 \leq L.\|\theta_t^a - \theta_t^b\|^2 \tag{111}$$

$$\leq L.\mu\left|\mathcal{L}_k(\theta_t^a) - \mathcal{L}_k(\theta_t^b)\right| \tag{112}$$

$$\leq \mathcal{O}\left(d \cdot W^D \cdot \sqrt{\frac{\log\frac{2}{\delta}}{n}}\right). \tag{113}$$

Thus we have with probability at least $1-\delta$

$$\|\hat{s}_t(x,t_k) - s_t^b(x,t_k)\|^2 \leq 2\|\hat{s}_{t_k}(x,t_k) - s_{t_k}^a(x,t_k)\| + 2.\|s_{t_k}^a(x,t_k) - s_{t_k}^b(x,t_k)\| \tag{114}$$

$$\leq \mathcal{O}\left(\log\left(\frac{1}{n}\right)\right) + \mathcal{O}\left(d \cdot \sqrt{\frac{\log\frac{2}{\delta}}{n}}\right). \tag{115}$$

$$\leq \mathcal{O}\left(d \cdot W^D \cdot \sqrt{\frac{\log\frac{2}{\delta}}{n}}\right). \tag{116}$$

$$\tag{117}$$

Taking expectation with respect to $x \sim p_{t_k}$ on both sides completes the proof. $\qquad\square$

## D  Intermediate Lemmas

**Lemma 4** (TV bound via Girsanov for reverse diffusions). *Let $X$ and $\tilde{X}$ on $[0,T]$ solve*

$$dX_t = \big(f(X_t,t) - \sigma^2(t)s_\star(X_t,t)\big)dt + \sigma(t)\,d\bar{W}_t, \qquad d\tilde{X}_t = \big(f(\tilde{X}_t,t) - \sigma^2(t)s_\theta(\tilde{X}_t,t)\big)dt + \sigma(t)\,d\bar{W}_t,$$

*with the same nondegenerate diffusion $\sigma(t) \in \mathbb{R}^{d\times d}$ (invertible for a.e. $t$) and the same initial law at time $T$. Let $\mathbb{P}$ and $\mathbb{Q}$ be the path measures of $X$ and $\tilde{X}$ on $C([0,T],\mathbb{R}^d)$. Assume Novikov's condition*

$$\mathbb{E}_{\mathbb{Q}}\exp\Big(\tfrac{1}{2}\int_0^T\|\sigma(t)(s_\theta(\tilde{X}_t,t) - s_\star(\tilde{X}_t,t))\|_2^2\,dt\Big) < \infty.$$

*Then*

$$\mathrm{TV}(\mathbb{P}, \mathbb{Q}) \le \frac{1}{2} \left( \mathbb{E}_{\mathbb{Q}} \int_0^T \left\| \sigma(t) \big( s_\theta(\tilde{X}_t, t) - s_\star(\tilde{X}_t, t) \big) \right\|_2^2 dt \right)^{1/2}.$$

*Proof.* Write the drift difference as

$$\Delta b(x, t) = -\sigma^2(t) \big( s_\star(x, t) - s_\theta(x, t) \big).$$

By Girsanov's theorem (under the stated Novikov condition), $\mathbb{P} \ll \mathbb{Q}$ and the Radon–Nikodym derivative is the exponential martingale driven by $u_t = \sigma(t)^{-1} \Delta b(\tilde{X}_t, t) = \sigma(t) \big( s_\theta(\tilde{X}_t, t) - s_\star(\tilde{X}_t, t) \big)$. The Cameron–Martin formula yields

$$\mathrm{KL}(\mathbb{P} \| \mathbb{Q}) = \frac{1}{2} \mathbb{E}_{\mathbb{Q}} \left[ \int_0^T \| u_t \|_2^2 \, dt \right] = \frac{1}{2} \mathbb{E}_{\mathbb{Q}} \left[ \int_0^T \left\| \sigma(t) \big( s_\theta(\tilde{X}_t, t) - s_\star(\tilde{X}_t, t) \big) \right\|_2^2 dt \right].$$

Applying Pinsker's inequality $\mathrm{TV}(\mathbb{P}, \mathbb{Q}) \le \sqrt{\mathrm{KL}(\mathbb{P} \| \mathbb{Q})/2}$ gives

$$\mathrm{TV}(\mathbb{P}, \mathbb{Q}) \le \frac{1}{2} \left( \mathbb{E}_{\mathbb{Q}} \int_0^T \left\| \sigma(t) \big( s_\theta - s_\star \big) \right\|_2^2 dt \right)^{1/2}.$$

Finally, the evaluation map $C([0, T], \mathbb{R}^d) \to \mathbb{R}^d$, $\omega \mapsto \omega(0)$, is measurable, so by data processing for $f$-divergences, $\mathrm{TV}(\mathcal{L}(X_0), \mathcal{L}(\tilde{X}_0)) \le \mathrm{TV}(\mathbb{P}, \mathbb{Q})$. □

Let $\{x_k\}_{k=0}^N$ and $\{\tilde{x}_k\}_{k=0}^N$ be Euler schemes with the same Gaussian noises,

$$x_{k-1} = x_k + \big( f_k - \sigma_k^2 s_\star(x_k, t_k) \big) \Delta t_k + \sigma_k \sqrt{\Delta t_k} \, \xi_k, \quad \tilde{x}_{k-1} = \tilde{x}_k + \big( f_k - \sigma_k^2 s_\theta(\tilde{x}_k, t_k) \big) \Delta t_k + \sigma_k \sqrt{\Delta t_k} \, \xi_k,$$

$\xi_k \sim \mathcal{N}(0, I)$ i.i.d. Then, with "traj" denoting trajectory measures,

$$\mathrm{KL}(\mathrm{traj}_\star \| \mathrm{traj}_\theta) = \frac{1}{2} \sum_{k=1}^N \mathbb{E} \left[ \left\| \sigma_k \big( s_\theta(x_k, t_k) - s_\star(x_k, t_k) \big) \right\|_2^2 \Delta t_k \right],$$

and hence by Pinsker's inequality we get,

$$\mathrm{TV}(\mathrm{traj}_\star, \mathrm{traj}_\theta) \le \frac{1}{2} \left( \sum_{k=1}^N \mathbb{E} \left\| \sigma_k \big( s_\theta - s_\star \big) \right\|_2^2 \Delta t_k \right)^{1/2}.$$

**Lemma 5** (Theorem 26.5 of (Shalev-Shwartz & Ben-David, 2014)). *Consider data $z \in Z$, the parametrized hypothesis class $h_\theta, \theta \in \Theta$, and the loss function $\ell(h, z) : \mathbb{R}^d \to \mathbb{R}$, where $|\ell(h, z)| \le c$. We also define the following terms*

$$L_D(h) = \mathbb{E}\ell(h, z) \tag{118}$$

$$L_S(h) = \frac{1}{m} \sum_{z_i \in \mathcal{S}} \ell(h, z_i) \tag{119}$$

*which denote the expected and empirical loss functions respectively.*

*Then,*

*With probability of at least $1 - \delta$, for all $h \in \mathcal{H}$,*

$$L_D(h) - L_S(h) \leq 2R(\ell \circ \Theta \circ S) + 4c\sqrt{\frac{2\ln(4/\delta)}{m}}. \tag{120}$$

*where $2R(\ell \circ \Theta \circ S)$ denotes the empirical Radamacher complexity over the loss function $\ell$, hypothesis parameter set $\Theta$ and the dataset $\mathcal{S}$*

**Lemma 6** (Extewnsion of Massart's Lemma (Bousquet et al., 2003))**.** *Let $\Theta''$ be a finite function class. Then, for any $\theta \in \Theta''$, we have*

$$\mathbb{E}_\sigma\left[\max_{\theta \in \Theta''} \sum_{i=1}^n f(\theta)\sigma_i\right] \leq ||f(\theta)||_2 \leq (BW)^L\left(d + \frac{L}{W}\right) \tag{121}$$

*where $\sigma_i$ are i.i.d random variables such that $\mathbb{P}(\sigma_i = 1) = \mathbb{P}(\sigma_i = -1) = \frac{1}{2}$. We get the second inequality by denoting $L$ as the number of layers in the neural network, $W$ and $B$ a constant such all parameters of the neural network upper bounded by $B$.*

*Proof.* Let $h_0 = x$, and for $\ell = 0, \ldots, L-1$ define the layer recursion

$$h_{\ell+1} = \sigma(W_\ell h_\ell + b_\ell),$$

where $W_\ell \in \mathbb{R}^{n_{\ell+1} \times n_\ell}$, $b_\ell \in \mathbb{R}^{n_{\ell+1}}$, and $n_\ell \leq W$ for hidden layers. We work with the $\ell_\infty$ operator norm:

$$\|W_\ell\|_\infty = \max_i \sum_j |(W_\ell)_{ij}| \ \leq \ B\,n_\ell \ \leq \ BW = \alpha.$$

Since $\sigma$ is 1-Lipschitz with $\sigma(0) = 0$, we have $\|\sigma(u)\|_\infty \leq \|u\|_\infty$ and thus

$$\|h_{\ell+1}\|_\infty \ \leq \ \|W_\ell\|_\infty \|h_\ell\|_\infty + \|b_\ell\|_\infty \ \leq \ \alpha \|h_\ell\|_\infty + B.$$

With $\|h_0\|_\infty \leq d$, iterating this affine recursion yields the standard geometric-series bound

$$\|h_L\|_\infty \ \leq \ \alpha^L d \ + \ B\sum_{i=0}^{L-1} \alpha^i \ = \ \alpha^L d \ + \ B\frac{\alpha^L - 1}{\alpha - 1} \quad (\alpha \neq 1),$$

and for $\alpha = 1$, $\|h_L\|_\infty \leq d + BL$. The scalar output $f(x)$ is either a coordinate of $h_L$ or obtained by applying the same 1-Lipschitz activation to a linear form of $h_L$; in either case, $|f(x)| \leq \|h_L\|_\infty$, giving the stated bound.

For the $\alpha \geq 1$ simplification, use $\sum_{i=0}^{L-1} \alpha^i \leq L\alpha^{L-1}$ to obtain

$$|f(x)| \ \leq \ \alpha^L d + BL\alpha^{L-1} \ = \ (BW)^L\left(d + \frac{L}{W}\right).$$

For $\alpha < 1$, since $\alpha^i \leq 1$, $\sum_{i=0}^{L-1} \alpha^i \leq L$ and hence $|f(x)| \leq d + BL$. Finally, substituting $W = S/L$ gives the size-based form

$$|f(x)| \ \leq \ \left(BS/L\right)^L\left(d + \frac{L^2}{S}\right).$$

$\square$

**Lemma 7** (Second Moment of a Symmetrically Truncated Normal)**.** *Let $X \sim \mathcal{N}(\mu, \sigma^2)$, and let $a > 0$. Then the second moment of $X$ conditioned on being outside the symmetric interval $[\mu - a, \mu + a]$ is given by*

$$\mathbb{E}[X^2 \mid |X - \mu| > a] = \mu^2 + \sigma^2 + \sigma a \cdot \frac{\phi\left(\frac{a}{\sigma}\right)}{1 - \Phi\left(\frac{a}{\sigma}\right)},$$

*where $\phi(z) = \frac{1}{\sqrt{2\pi}}e^{-z^2/2}$ is the standard normal probability density function (PDF), and $\Phi(z)$ is the standard normal cumulative distribution function (CDF).*

*Proof.* Let $X \sim \mathcal{N}(\mu, \sigma^2)$. We aim to compute the second moment of $X$ conditioned on the event that it lies outside an interval centered at its mean

$$\mathbb{E}[X^2 \mid |X - \mu| > a]$$

This represents the expected squared value of $X$, given that $X$ is in the tails of the distribution (i.e., more than $a$ units away from the mean).

By definition, the conditional expectation is

$$\mathbb{E}[X^2 \mid |X - \mu| > a] = \frac{\mathbb{E}[X^2 \cdot \mathbf{1}_{\{|X-\mu|>a\}}]}{\mathbb{P}(|X - \mu| > a)}$$

The numerator integrates $X^2$ over the tail regions $(-\infty, \mu - a) \cup (\mu + a, \infty)$, while the denominator is the probability mass in those same regions.

To simplify the integrals, we standardize $X$. Define the standard normal variable

$$Z = \frac{X - \mu}{\sigma} \sim \mathcal{N}(0, 1) \quad \Rightarrow \quad X = \mu + \sigma Z$$

Define $\alpha = \frac{a}{\sigma}$. Then

$$|X - \mu| > a \quad \Leftrightarrow \quad |Z| > \alpha$$

Our conditional second moment becomes

$$\mathbb{E}[X^2 \mid |X - \mu| > a] = \mathbb{E}[(\mu + \sigma Z)^2 \mid |Z| > \alpha]$$

Expanding the square inside the expectation

$$(\mu + \sigma Z)^2 = \mu^2 + 2\mu\sigma Z + \sigma^2 Z^2$$

Taking the conditional expectation

$$\mathbb{E}[(\mu + \sigma Z)^2 \mid |Z| > \alpha] = \mu^2 + 2\mu\sigma \mathbb{E}[Z \mid |Z| > \alpha] + \sigma^2 \mathbb{E}[Z^2 \mid |Z| > \alpha]$$

Since the standard normal distribution is symmetric and the region $|Z| > \alpha$ is also symmetric, we have

$$\mathbb{E}[Z \mid |Z| > \alpha] = 0$$

Thus, the expression simplifies to

$$\mathbb{E}[X^2 \mid |X - \mu| > a] = \mu^2 + \sigma^2 \mathbb{E}[Z^2 \mid |Z| > \alpha]$$

By definition

$$\mathbb{E}[Z^2 \mid |Z| > \alpha] = \frac{\int_{|z|>\alpha} z^2 \phi(z)\, dz}{\mathbb{P}(|Z| > \alpha)} = \frac{2 \int_{\alpha}^{\infty} z^2 \phi(z)\, dz}{2(1 - \Phi(\alpha))} = \frac{\int_{\alpha}^{\infty} z^2 \phi(z)\, dz}{1 - \Phi(\alpha)}$$

Using Intergration by Parts we get,

$$\int_{\alpha}^{\infty} z^2 \phi(z)\, dz = \phi(\alpha)\alpha + 1 - \Phi(\alpha)$$

Therefore

$$\mathbb{E}[Z^2 \mid |Z| > \alpha] = \frac{\phi(\alpha)\alpha + 1 - \Phi(\alpha)}{1 - \Phi(\alpha)} = 1 + \frac{\alpha\phi(\alpha)}{1 - \Phi(\alpha)}$$

Substitute back into the expression for $\mathbb{E}[X^2 \mid |X - \mu| > a]$

$$\mathbb{E}[X^2 \mid |X - \mu| > a] = \mu^2 + \sigma^2 \left(1 + \frac{\alpha\phi(\alpha)}{1 - \Phi(\alpha)}\right)$$

Recall that $\alpha = \frac{a}{\sigma}$, so the final expression becomes

$$\mathbb{E}[X^2 \mid |X - \mu| > a] = \mu^2 + \sigma^2 + \sigma a \cdot \frac{\phi\left(\frac{a}{\sigma}\right)}{1 - \Phi\left(\frac{a}{\sigma}\right)}$$

$\square$

**Lemma 8** (Linear Growth of Finite Neural Networks). *Let $f_\theta : \mathbb{R}^d \to \mathbb{R}$ be the output of a feedforward neural network with a finite number of layers and parameters and $\theta \in \Theta$ where $\Theta$ has a finite number of elements. Suppose that each activation function $\sigma : \mathbb{R} \to \mathbb{R}$ satisfies the growth condition*

$$|\sigma(z)| \leq A + B|z|, \quad \text{for all } z \in \mathbb{R},$$

*for constants $A, B \geq 0$. Then there exists a constant $C_\Theta > 0$ such that for all $x \in \mathbb{R}^d$,*

$$|f(x)| \leq C_\Theta(1 + \|x\|).$$

*Proof.* We proceed by induction on the number of layers in the network.

**Base case: One-layer network.** Let the network be a single-layer function

$$f(x) = \sum_{i=1}^{k} a_i\, \sigma(w_i^\top x + b_i),$$

where $w_i \in \mathbb{R}^d$, $b_i \in \mathbb{R}$, and $a_i \in \mathbb{R}$. Then

$$|f(x)| \leq \sum_{i=1}^{k} |a_i| \cdot |\sigma(w_i^\top x + b_i)|.$$

Using the growth condition on $\sigma$, we get

$$|\sigma(w_i^\top x + b_i)| \leq A + B|w_i^\top x + b_i| \leq A + B(\|w_i\|\|x\| + |b_i|).$$

Hence

$$|f(x)| \leq \sum_{i=1}^{k} |a_i| \left(A + B(\|w_i\|\|x\| + |b_i|)\right) = C_0 + C_1\|x\|,$$

where $C_0, C_1$ are constants depending only on the network parameters. Therefore

$$|f(x)| \leq C(1 + \|x\|) \quad \text{with } C = \max\{C_0, C_1\}.$$

**Inductive step.** Assume the result holds for all networks with $L$ layers, i.e., for any such network $f_L(x)$,

$$|f_L(x)| \leq C_L(1 + \|x\|).$$

Now consider a network with $L + 1$ layers, defined by

$$f_{L+1}(x) = \sum_{j=1}^{k} a_j \, \sigma(f_L^{(j)}(x)),$$

where each $f_L^{(j)}(x)$ is an output of a depth-$L$ subnetwork. By the inductive hypothesis

$$|f_L^{(j)}(x)| \leq C_j(1 + \|x\|).$$

Applying the activation bound

$$|\sigma(f_L^{(j)}(x))| \leq A + B|f_L^{(j)}(x)| \leq A + BC_j(1 + \|x\|).$$

Then

$$|f_{L+1}(x)| \leq \sum_{j=1}^{k} |a_j| \cdot |\sigma(f_L^{(j)}(x))| \leq \sum_{j=1}^{k} |a_j|(A + BC_j(1 + \|x\|)) = C_{L+1}(1 + \|x\|),$$

for some constant $C_{L+1} > 0$. This completes the induction.

## Examples of Valid Activation Functions

The condition $|\sigma(z)| \leq A + B|z|$ holds for most common activations

- **ReLU**: $\sigma(z) = \max(0, z) \Rightarrow |\sigma(z)| \leq |z|$

- **Leaky ReLU**: bounded by linear function of $|z|$

- **Tanh**: bounded by $1 \Rightarrow A = 1, B = 0$

- **Sigmoid**: bounded by 1

$\square$

