# OpenReview forum: "Improved Sample Complexity Bounds For Diffusion Model Training Without Empirical Risk Minimizer Access"
_TMLR — Accepted by TMLR_

### Review · Reviewer_3dyA · 2025-12-17

**Summary Of Contributions:**

This work essentially provides a theoretical analysis of the error committed by diffusion models trained with score estimation. To my understanding, the main contribution of this work is to provide a similar bound to previous works regarding sample complexity ($\mathcal{O}(\epsilon^{-4})$ instead of the $\mathcal{O}(\epsilon^{-5})$ of previous works, where $\epsilon$ is the total variation between real and estimated distributions) while relaxing the set of assumptions made. Most importantly, the authors only need to assume that the data distribution has bounded variance (instead of bounded domain) and do not need to assume access to a minimizer of the empirical risk. The paper is generally well-written and the proofs well explained (I have only looked at some of them, I didn't proof-read everything). While I miss some empirical confirmation of the theoretical results, these represent a step towards more realistic set of assumptions to what it was previously known and therefore a worthy contribution to the field.

**Additional Comments:**

_Disclaimer:_ This is not my field and I am not familiar with the literature. My review is based on what it is shared by the authors and the limited literature review I have done during this review. I have looked superficially at some of the proofs but I have certainly not proof-read the appendices.

**Audience:**

Yes

**Audience Explanation:**

Yes, diffusion models are still a hot topic in ML research and the theoretical understanding on the number of samples required to get a good approximation is of interest to the community.

**Broader Impact Concerns:**

I do not think that there are any ethical concerns to consider in this particular work.

**Claims And Evidence:**

Yes

**Claims Explanation:**

As I said above, the paper is in general really clearly written and even an outsider to the field like me can follow it up without many issues. While I would need to read every demonstration in details, the ones I have looked seem rather simple (in a good way) and clearly explained, which encourage me to lean towards trusting the theoretical results presented in the main pages of this submission.

Once again, I think the paper would greatly benefit of a small but convincing experiment (or existing results taken from previous works if computational budget is an issue) showing that indeed the sample complexity bound given in this paper holds in practice.

**Requested Changes:**

I would request from the authors to make a deeper pass on the manuscript to polish notation and make it consistent across all pages (maybe using LaTeX macros?), as it can become distracting. For one example, in Eq. 18 it is defined as $A(k)$ but literally in the next equation the same is written as $A_k$. Similarly, I believe the dots in Eq. 25 and others are meant to be $\cdot$ (`\cdot`). Or in the problem formulation, it is defined $n_k$ but used $n$ right after.

Moreover, I believe the $dx$ in Eq. 18 should be removed and the $(t_{k+1} - t_k)$ is wrongly duplicated in Eq. 19.
In the same spirit, I could spot some typos ("establishes" instead of "establish" in the related work, or "so that analyzes" instead of "analyses").

Finally, there are some notation and terms that would required further clarification. For example, $D$ is never introduced in the main text, and in the proof of Lemma 4 I would appreciate it if there were some references to the 3 different theorem and conditions thrown in the same paragraph.

Regarding some empirical results, I think they would strengthen the paper significantly, but I do not consider them necessary for acceptance.

---

> ### Author Response · Authors · 2026-01-13
>
> We thank the reviewer for the positive assessment of the technical contribution and for highlighting concrete ways to improve clarity and polish. We especially appreciate the detailed notes on notation inconsistencies and typos; we agree these can be distracting and we have addressed the issues pointed out by the reviewer carefully in the revision. Below we respond point-by-point to the requested changes and describe specific edits we will make.
>
> ---
>
> >### Notation polishing and consistency (including Eq. 18–19, Eq. 25, and the problem formulation)
>
> **Reviewer comment.** Make a deeper pass to polish notation and make it consistent across the manuscript (e.g., Eq. 18 vs the next equation, dots in Eq. 25, and minor inconsistencies immediately after definitions in the problem formulation).
>
> **Response.** We agree. We have conducted a full notation pass and standardized the notation to ensure consistency across the main text and appendices.
>
> **Concrete fixes (as flagged by the reviewer).**
> -**Regarding Grammatical Typos** We have done a pass of the manuscipt to correct the typos pointed out by the reviewer and ensure no other ones are present.
> - **Incorrect usage of n instead of n_{k}** We ensured that the sample size is denoted by $n_{k}$ ajnd not $n$ throughout the text.
> - **Remove spurious or duplicated factors.** We have removed the $dx$ in Eq. 18 and $(t_{k+1}-t_{k})$ is repeated and has been removed.
> - **Eq. 25 and other “dot” notations.** We replaced the informal dot usage with standard notation
>
> - **Problem formulation definition/use mismatch.** We have fixed the inconsistencies in the revision (including consistent use of subscripts/superscripts for early-stopped, discretized, and estimated distributions).
>
> - **With regards to D** $D$ is the depth of the neural network and has already been introduced in the section titled **Problem Formulation** on page 4.

---

> > ### Comment · Reviewer_3dyA · 2026-01-22
> >
> > Dear authors,
> >
> > Apologies for the late response, but there are no further questions from my side. Thank you once again for the rebuttal!

---

> > > ### Author Response · Authors · 2026-01-22
> > >
> > > Thank you for the update. We are pleased that our response helped address your comments, and we appreciate your time and thoughtful review.

---

### Review · Reviewer_uAqs · 2026-01-05

**Summary Of Contributions:**

The contributions of the paper are threefold:
- Provides a finite-sample sample complexity bound for diffusion/DDPM training via score estimation that targets TV error between the generated distribution and the (early-stopped) forward-noised data distribution, improving the reported scaling to $\mathcal{O}\left(\epsilon^{-4}\right)$ (plus approximation error) without assuming access to an empirical risk minimizer (ERM).
- Introduces a three-way decomposition of score estimation error into approximation, statistical, and optimization terms, and claims to bound each under (i) bounded second moment of data, (ii) a PL condition for the score loss, and (iii) smoothness + bounded gradient noise for SGD;
- Re-analyzes prior work (notably Gupta et al.) and argues their claimed $\mathcal{O}\left(\epsilon^{-3}\right)$ effectively becomes $\mathcal{O}\left(\epsilon^{-5}\right)$ after accounting for accumulation/union bounds across discretization steps.

**Additional Comments:**

The distinction between bounding $\mathrm{TV}(\hat p_{t_0}, p_{t_0})$ and $\mathrm{TV}(\hat p_{t_0}, p_0)$ is important and is handled appropriately. However, the paper in general could perhaps benefit from a bit clearer exposition, more consistent notation, and tighter bookkeeping of constants and parameter dependencies. Improving readability, especially in the appendices, would help verifying the technical claims.

**Audience:**

Yes

**Audience Explanation:**

The paper addresses questions that are directly relevant to the theory-oriented portion of the TMLR audience, particularly researchers working on diffusion models, non-asymptotic generalization bounds, and optimization-aware learning theory. All topics that are of extreme interest right now. The attempt to remove the empirical risk minimization assumption and to analyze finite-time stochastic gradient descent in terms of total variation distance is timely and of clear theoretical interest.

**Broader Impact Concerns:**

This work is purely theoretical and does not introduce new modeling capabilities or deployment mechanisms. As such, it has no immediate societal impact. Any broader impact is indirect and stems from improved theoretical understanding of diffusion models, which may contribute to both beneficial applications and general dual-use risks associated with generative models. So the paper itself does not raise specific new ethical or safety concerns.

**Claims And Evidence:**

Yes

**Claims Explanation:**

The claims appear to be supported, but I have some questions/comments before giving a positive answer to this question.

While the overall proof structure is reasonable, the main $\widetilde{O}(\varepsilon^{-4})$ sample complexity claim is not fully clear for me.

The reduction of total variation error to a weighted sum of per-time score errors,
$$
\sum_{k=1}^K \frac{E_k}{\sigma_k^2} \;\lesssim\; \varepsilon^2,
$$
is standard and sound. However, some key issues perhaps remain:

- Optimization error:
The optimization error bound
$$
E_k^{\mathrm{opt}} \;\lesssim\; W D d \sqrt{\frac{\log(1/\delta)}{n_k}}
$$
is derived under a PL condition but does not clearly account for the finite number of SGD iterations. Standard PL-based analyses depend explicitly on the iteration count $T$, which is not reflected in the final bounds.

- Parameter dependence:
The per-time error bounds scale as $W D d$, while the main theorem requires
$$
n_k = \Omega\!\left(
W^2 D \, d^2 \log\!\left(\frac{K}{\delta}\right)
\varepsilon^{-4}
\sigma_k^{-4}
\right).
$$
The apparent mismatch in depth dependence (missing a factor of $D$) is not explained.

- Statistical analysis and assumptions:
The statistical error argument relies on truncation and conditional Gaussianity but is difficult to verify as written. In addition, assuming a PL condition for each time-indexed loss $L_k(\theta)$ is strong for realistic diffusion models, and the approximation error $\varepsilon_{\mathrm{approx}}$ is left uncontrolled.

I would appreciate if the authors could clarify so that I can answer the question positively.

*Edit: after the authors response, I am confident and validating the correctness of the paper and updated the answer to the question above to "Yes". Thank you.

**Requested Changes:**

My requested changes are mostly very technical and relate to the first question, of correctness:
- Make the optimization analysis explicitly finite-time by stating the required number of SGD iterations per time step and incorporating this dependence into the final bounds.
- Reconcile the dependence on network width $W$, depth $D$, and dimension $d$ between the per-time error bounds and the main sample complexity result.
- Clarify and streamline the statistical error analysis, including precise definitions of truncation and a clear connection to the actual score-matching objective.
- Clearly state whether the theory assumes separate models for each time step or a single time-conditioned network, and explain how the results extend to the shared-network setting.
- Tighten the comparison with prior work by explicitly matching assumptions and carefully qualifying claims about improved rates.

It is relevant that some misunderstanding from my part is clarified or that some things are corrected / added of clarification.

---

> ### Author Response · Authors · 2026-01-13
>
> We thank the reviewer for the detailed and constructive feedback. We agree that several aspects of the presentation can be clarified, especially around (i) finite-time optimization, (ii) bookkeeping of parameter dependence in $(W,D,d)$ across intermediate lemmas and the main theorem, (iii) the statistical truncation argument, (iv) whether training is per-time-step or time-conditioned, and (v) comparison with prior work. Below we respond point-by-point and describe concrete revisions we will make.
>
> ---
>
>
>
> ## 1) Make the optimization analysis explicitly finite-time
>
> **Reviewer comment.** “Make the optimization analysis explicitly finite-time by stating the required number of SGD iterations per time step and incorporating this dependence into the final bounds.”
>
> **Response.** Thank you for pointing this out. we have added a section after the Optimization Lemma in the main text where we go over how the batch size at each step changes leading to the result in the optimization lemma.
>
>
>
> ---
>
> ## 2) Reconcile dependence on width $W$, depth $D$, and dimension $d$, Clarifying the reviewer’s “$WDd$” remark
>
> **Reviewer comment.** “Reconcile the dependence on network width $W$, depth $D$, and dimension $d$ between the per-time error bounds and the main sample complexity result.”
>
> **Reviewer comment.** “The per-time error bounds scale as $WDd$.”
>
> **Response.**  In our setting, the per-time estimation bound (for a depth-$D$, width-$W$ score network in $d$ dimensions) scales as
>
> $$
> \text{(per-time score error)} \sim W^{D}\cdot d \times \text{(concentration in }n_k\text{)},
> $$
>
> (See Eq.25). Accordingly, when we convert a per-time bound into a sample complexity requirement, we must choose $n_k$ large enough to make the per-time loss contribution smaller than $\varepsilon^2$ (more precisely, to ensure the per-time score estimation error term $A(k)$ is at most $\varepsilon^2/\sigma_k^2$), as carried out in Eq. (43). This squaring step leads to a required sample size scaling as
>
> $$
> n_k = \Omega\left(W^{2D}d^{2}\times \cdots \right),
> $$
>
> matching the sample complexity dependence stated in the theorem and in the choice of $n_k$.
>
> ---
>
> ## 3) Justification for the PL assumption
>
> We agree that assuming a Polyak–Łojasiewicz (PL) condition for each time-indexed loss \(L_k(\theta)\) is nontrivial. However, there is supporting theoretical evidence in closely related settings. Each per-time objective, $L_k(\theta) = E_{x\sim p_{t_k}} \| \|s_\theta(x,t_k) - \nabla \log p_{t_k}(x) \|\|^2$, is a **squared-error objective** over the network output. In several regimes relevant to modern neural networks, particularly **overparameterized** or **lazy-training (NTK-like)** settings, squared-loss objectives have been shown to satisfy PL (or PL-type) conditions, yielding **linear convergence for gradient descent** and **geometric decay for SGD up to a noise floor** in works such us Liu et al. (2022) . Since our score-matching loss at each $(t_k)$ has the structure of a regression problem in $(\mathbb{R}^d)$, these results provide evidence that a PL assumption can be appropriate for the loss class we consider (under standard width/initialization/norm conditions).
>
> In the revised manuscript, we have moderated the presentation to avoid overselling. We will explicitly state that PL is not claimed to universally hold in practice, but rather serves as a minimal, standard global convergence-enabling assumption that allows us to replace the ERM oracle with a transparent, optimization-aware finite-time analysis.
>
> ### References
> Chaoyue Liu, Libin Zhu, and Mikhail Belkin. Loss landscapes and optimization in over-parameterized non-linear systems and neural networks. Applied and Computational Harmonic Analysis, 59:85–116, 2022.
>
> ---
> ## 4) Separate networks per time step vs. a shared time-conditioned network
>
> **Reviewer comment.** “Clearly state whether the theory assumes separate models for each time step or a single time-conditioned network, and explain how the results extend to the shared-network setting.”
>
> **Response.** For conceptual clarity, our main analysis considers **a separate score network for each discretized time step** $(t_k)$, trained on samples from \(p_{t_k}\). This aligns with the structure of the proof, which decomposes and bounds the error **per time index**, and matches the description that we “learn the score function at each $(t_k)$” using SGD.
>
> Our function class is written broadly enough to include a **single time-conditioned network** $s_\theta(x,t)$, mapping $(\mathbb{R}^d \times [0,T] \to \mathbb{R}^d)$, trained using the time-indexed loss  $L_k(\theta) = E_{x\sim p_{t_k}} \| \|s_\theta(x,t_k) - \nabla \log p_{t_k}(x) \|\|^2$.
> As it stands the analysis cannot be directly extended to the a shared network architecture. We leave that extension for future work.

---

> ### Author Response · Authors · 2026-01-13
>
> ## 5) Clarify and streamline the statistical error analysis (truncation and conditional Gaussianity)
>
> **Reviewer comment.** “Clarify and streamline the statistical error analysis, including precise definitions of truncation and a clear connection to the actual score-matching objective.”
>
> **Response.** We agree the statistical argument can be presented more cleanly. We have added a section in the main text that gives a proof outline of the statistical error.
>
> ---
> ## 6) Tighten comparison with prior work
>
> **Reviewer comment.** “Tighten the comparison with prior work by explicitly matching assumptions and carefully qualifying claims about improved rates.”
>
> **Response.** For Assumption 1 we have added text which described how prior works used the same\similiar assumptions. Note that assumptions 2-4 are only needed in our work since we have not assumed access to  the ERM which was done in all prior works. We have added text that points this out.

---

> ### Comment · Reviewer_uAqs · 2026-01-23
> **Response to authors**
>
> I thank the authors for their response. I am happy with the clarifications provided and modifications performed. I will update my review to reflect this. Thank you.

---

### Review · Reviewer_xBMb · 2026-01-09

**Summary Of Contributions:**

This paper studies the sample complexity of learning a diffusion model under a PL condition, which is claimed to be a weaker form of convexity, rather than access to an empirical risk minimizer (ERM) as assumed by previous works. Under this assumption, it achieves a $O(\epsilon^{-4})$  bound, which improves upon the $O(\epsilon^{-5})$ from previous works

**Audience:**

Yes

**Audience Explanation:**

The paper addresses a fundamental problem in diffusion models.

**Claims And Evidence:**

Yes

**Claims Explanation:**

This paper points out an error from Gupta et al (2024) which I appreciate -- it does seem that the paper reports the wrong bound in the simplified theorem in the main body of the paper -- however, it's also the case that if one just directly plugs in the bound on $K$
 stated in Theorem C.2 into the bound in equation (35), one recovers the correct $O(\epsilon^{-5})$
 bound. Moreover, the main contribution of Gupta et al (2024) was to improve on the dependence on the Wasserstein error and dependence on depth of the neural network -- the dependence on $\epsilon$ was not a focus.

I also like that this paper does attempt to circumvent the need for ERM access assumed in previous works. The improvement in the $\epsilon$ dependence as a result of the new PL assumption is interesting.

However, I have a number of concerns, highlighted in future responses.

**Requested Changes:**

The PL assumption still seems quite strong -- it only seems to apply to local convergence for overparameterized neural networks, which is still a very restrictive setting. In general, the paper suffers from overselling its contributions -- it's not as though the authors are able to remove the (admittedly unrealistic) ERM access assumption altogether -- it is simply replaced with a different unrealistic assumption, that likely also does not hold in practice.

The flaw pointed out in Gupta et al (2024) is not stated with appropriate context -- it's the case Theorem C.5 as stated is correct, the error only appears in the simplified theorem statement.

Moreover, unlike the polynomial dependence in Gupta et al (2024), this paper has an exponential dependence on the depth of the neural network.

There are several typos in the paper, and the presentation in general is quite poor. For instance, Assumption 1 is a second moment assumption on the data distribution, but immediately after, the authors say "In contrast, our analysis only requires the data distribution to be sub-Gaussian, making our results applicable to a significantly broader class of distributions." which is much stronger than a second moment assumption.

Overall, this paper has the potential to be a good contribution, but is not yet ready for publication in my opinion.

---

> ### Author Response · Authors · 2026-01-13
>
> >### Response regarding Gupta et al. (2024) and the $\mathcal{O}(\epsilon^{-5})$
>
> Thank you for the clarification. We agree that our discussion of Gupta et al. (2024) should be updated and contextualized more carefully. In particular, in the revised version of Gupta et al. (2024) on arXiv (updated version at arXiv:2311.13745v4, Nov 2025), the simplified theorem statement has been corrected, and therefore we have removed the claim that there is an error in their work. In the revised manuscript, we simply report the $\mathcal{O}(\epsilon^{-5})$ dependence as the correct baseline implied by their current results.
>
> We have also revised the comparison section to better reflect the intent and strengths of Gupta et al. (2024). In particular, we have explicitly acknowledge that their primary contribution lies in achieving improved dependence on neural network parameters (e.g., depth and architectural complexity). Our revised discussion therefore (i) accurately presents the  $\mathcal{O}(\epsilon^{-5})$rate in Gupta et al. (2024), and (ii) highlight that Gupta et al. (2024) enjoys more favorable dependence on network depth and related parameters than our current bounds.
>
>
> >### Response regarding the PL assumption versus ERM access
>
> We appreciate the concern that replacing an ERM-access assumption with a Polyak–Łojasiewicz (PL) assumption may appear to trade one unrealistic condition for another. We have clarified the role of this assumption more carefully in the revision.
>
> First, the PL inequality is strictly weaker than strong convexity and does not require convexity. It is an optimization regularity condition that can hold for nonconvex objectives and is widely used to establish global convergence guarantees for first-order methods when it holds globally.
>
> Second, within our analysis framework, PL is essentially the weakest standard assumption that yields global convergence rates strong enough to close a finite-sample optimization analysis. Our goal is to remove the assumption of access to an empirical risk minimizer and instead provide an explicit, finite-time SGD guarantee that appears as an optimization error term in the three-way decomposition. To achieve this, some form of global error–gradient relationship is required. Among commonly used assumptions, the PL inequality is a canonical and comparatively mild choice: it is compatible with nonconvex landscapes and is significantly weaker than convexity, while still enabling global convergence guarantees for SGD.
>
> In the revised manuscript, we have moderated the presentation to avoid overselling. We will explicitly state that PL is not claimed to universally hold in practice, but rather serves as a minimal, standard global convergence-enabling assumption that allows us to replace the ERM oracle with a transparent, optimization-aware finite-time analysis.

---

### Decision · Action_Editor_GiTs · 2026-04-03

**Recommendation:** Accept as is

**Audience:**

Yes

**Audience Explanation:**

This paper studies the theory of diffusion, a relevant topic to TMLR. It contributes with new theoretical insights, and I believe it is of interest to the theoretical part of the TMLR community.

**Claims And Evidence:**

Yes

**Claims Explanation:**

This submission studies the sample complexity of diffusion models under relaxed assumptions, removing the need for empirical risk minimisation and providing a refined theoretical analysis with improved rates. The reviewers unanimously agree that the work is technically sound, relevant to the TMLR theory community, and offers a meaningful contribution to the understanding of diffusion models.

While initial concerns were raised regarding the strength of assumptions, clarity of exposition, and certain technical assumptions, these issues were addressed during the discussion, and all reviewers were happy with the author's rebuttal. In particular, the paper’s contribution in eliminating the ERM assumption and providing a structured error decomposition represents a clear advance over prior work.

In view of the unanimous positive assessment and the satisfactory resolution of all concerns, I recommend the acceptance of this paper to TMLR.